# Diurnal carbon monoxide observed from a geostationary infrared hyperspectral sounder: First result from GIIRS onboard FengYun-4B

Zhao-Cheng Zeng[1], Lu Lee[2], Chengli Qi[2]

[1]School of Earth and Space Sciences, Peking University, Beijing 100871, China
[2]Innovation Center for FengYun Meteorological Satellite, Key Laboratory of Radiometric Calibration and Validation for Environmental Satellites, National Satellite Meteorological Center, China Meteorological Administration, Beijing 100081, China

*Correspondence to*: Z.-C. Zeng (zczeng@pku.edu.cn)

**Abstract.** The Geostationary Interferometric Infrared Sounder (GIIRS) onboard FengYun-4 series satellites is the world's first geostationary hyperspectral infrared sounder. With hyperspectral measurement collected from a geostationary orbit covering the carbon monoxide (CO) absorption window around 2150 cm$^{-1}$, GIIRS provides a unique opportunity for monitoring the diurnal variabilities of atmospheric CO over East Asia. In this study, we develop the FengYun Geostationary satellite Atmospheric Infrared Retrieval (FY-GeoAIR) algorithm to retrieve the CO profiles using observations from GIIRS onboard FY-4B, which was launched in June 2021, and provide CO maps at a spatial resolution of 12 km and a temporal resolution of 2 hours. The performance of the algorithm is first evaluated by conducting retrieval experiments using simulated synthetic spectra. The result shows that the GIIRS data provide significant information for constraining CO profiles. The degree of freedom for signal (DOFS) and retrieval error are both highly correlated with thermal contrast (TC), the temperature difference between the surface and the lower atmosphere. Retrieval results from one month of GIIRS spectra in July 2022 show that the DOFS for the majority is between 0.8 and 1.5 for the CO total column and between 0 and 0.8 for the bottom 3-layer ranging from the surface to 3 km a.s.l.. Consistent with CO retrievals from low-earth-orbit (LEO) infrared sounders, the largest observation sensitivity, as quantified by the averaging kernel (AK), is in the free troposphere at around 3-6 km. The diurnal changes in DOFS and vertical sensitivity of observation are primarily driven by the diurnal TC variabilities. Finally, we compare the CO total columns between GIIRS and IASI and find that the two datasets show good consistency in capturing the spatial and temporal variabilities. This study demonstrates that the GIIRS retrievals are able to reproduce the temporal variability of CO total columns over East Asia in the daytime in July. Nevertheless, the retrievals have low detectivity in the nighttime due to their weak sensitivity to the ground level CO changes limited by low information content. Model assimilation that takes into account the retrieved diurnal CO profiles and the associated vertical sensitivity will have potential in improving local and global air quality and climate research over East Asia.

# 1 Introduction

Observing atmospheric composition from space provides critical data for forecasting air quality, assessing climate change, and monitoring the long-term variabilities in tropospheric and stratospheric compositions. In the last two decades, satellite-borne instruments onboard polar-orbiting satellites in Low-Earth Orbit (LEO) have demonstrated their full capabilities in observing the atmospheric composition (**e.g., Clerbaux et al., 2003; Crevoisier et al., 2014**; **Shephard et al., 2015; Buchwitz et al., 2005; Borsdorff et al., 2018**). However, a single LEO satellite has a revisit time of 12 hours over the equator. In general, only one (for near-infrared or UV–visible instrument) or two (for thermal infrared sounder) observations are available each day for the same spot. Critical information on the diurnal cycle of atmospheric composition is, however, not available from LEO satellites. As an important advancement over current LEO instruments, measurements from geostationary (GEO) orbit can provide contiguous coverage with similar or higher spatial resolution and a revisit time of 1-2 hours, which would provide breakthrough measurements for numerical weather prediction and support high-temporal-resolution air quality forecasting (**Schmit et al., 2009**).

The Geostationary Interferometric Infrared Sounder (GIIRS) onboard FengYun-4 series satellites, launched in 2016 (FY-4A) and 2021 (FY-4B), respectively, is the world's first geostationary hyperspectral infrared sounder (**Yang et al., 2017**). With a spectral resolution of 0.625 cm$^{-1}$, similar to current LEO satellites, GIIRS provides a unique opportunity for observing the diurnal variabilities of atmospheric composition over East Asia, as has been demonstrated in retrieving atmospheric ammonia (**Clarisse et al., 2021**). Existing on-orbit GEO instruments for observing air quality also include the Geostationary Environment Monitoring Spectrometer (GEMS) by South Korea which was launched in Feb. 2020. GEMS was designed to measure air quality in Asia using ultraviolet and visible (UV/VIS) bands (**Kim et al., 2020**). Future GEO missions with hyperspectral capabilities include ESA's Sentinel-4 mission onboard the Meteosat Third Generation Sounder platform, which is made up of the thermal Infrared Sounder (IRS) for providing profiles of temperature and humidity and the Ultraviolet Visible Near-infrared (UVN) spectrometer for monitoring air quality trace gases and aerosols in Europe (**Ingmann et al., 2012; Holmlund et al., 2021**), and NASA's Tropospheric Emissions: Monitoring of Pollution (TEMPO; **Zoogman et al., 2017**) that will track air quality in North America. In addition, the to-be-launched greenhouse gas targeted mission: Geostationary Carbon Cycle Observatory (GeoCarb) by NASA was designed to measure carbon dioxide ($CO_2$), methane ($CH_4$), and carbon monoxide (CO) throughout the Americas (**Polonsky et al., 2014**).

As an important trace gas for understanding air quality and climate forcing, CO is a direct product of incomplete combustion primarily from biomass and fossil fuel on the surface and a by-product of oxidation of $CH_4$ and non-methane hydrocarbons in the atmosphere (**Brenninkmeijer and Novelli, 2003**). Being a precursor to the formation of tropospheric ozone, CO also plays an important role in tropospheric chemistry (**Chin et al., 1994**). Because of its low background concentration and moderately long lifetime (weeks to months) in the troposphere, CO is an effective tracer for the long-range transport of pollution (**Forster et al., 2001**) and carbon emissions (**Gamnitzer et al., 2006**). Nadir observation of CO from space has been providing long-term global coverage from both thermal (TIR) and near-infrared (NIR) instruments. One of the earliest attempts to retrieve

atmospheric CO was made by the Interferometric Monitor of Greenhouse gases (IMG) onboard the Japanese ADEOS satellite (**Barret et al., 2005**). From the early 2000s, the Measurements Of Pollution in The Troposphere (MOPITT) instrument onboard

NASA's Terra satellite launched was the first to provide routine global maps of CO daily (**Deeter et al., 2003**). Following missions with CO nadir observation capability includes the Infrared Atmospheric Sounding Interferometer (IASI) onboard Metop-A/B (**Hurtmans et al., 2012**), the Scanning Imaging Absorption Spectrometer for Atmospheric Chartography (SCIAMACHY) onboard the European ENVISAT satellite (**Buchwitz et al., 2005**), the Tropospheric Emission Sounder (TES) onboard NASA's Aura satellite (**Luo et al., 2007**), and the Cross-track Infrared Sounder (CrIS) onboard the Suomi National

Polar-orbiting Partnership platform (**Goldberg et al., 2013; Gambacorta et al., 2014**). More recently, TROPOMI and GOSAT-2, covering the NIR spectra, provides additional daily global views of CO (**Borsdorff et al., 2018; Noël et al., 2022**). However, none of the current instruments and missions provide diurnal CO measurements with high temporal resolution from a GEO platform.

In this study, we report the first result of diurnal CO retrieved from the hyperspectral infrared measurements by the GIIRS

using the Feng-Yun Geostationary satellite Atmospheric Infrared Retrieval (FY-GeoAIR) algorithm. The retrieval algorithm uses the absorption feature of CO's fundamental 1-0 rotation-vibration band centered around 4.7 μm (2150 cm$^{-1}$), which allows the measurement to be made during the daytime and the nighttime and provides important vertical information from the retrieval (**Crevoisier, 2018**). The clear-sky CO retrievals and uncertainties are produced as well as the averaging kernel (AK) matrix for each retrieval that quantifies its vertical observation sensitivity and information content.

This paper is organized as follows. In **Sect. 2**, the GIIRS instrument and the observed spectra data are introduced. In **Sect. 3** and **Sect. 4**, we describe the details of the forward model based on radiative transfer and the inverse model based on optimal estimation theory, respectively. We show results from a simulation experiment in **Sect. 5** to assess the performance of the retrieval algorithm. Results of CO retrievals from applying the algorithm to GIIRS spectra in July of 2022 are demonstrated in **Sect. 6**, followed by discussions and conclusions in **Sect. 7** and **Sect. 8**, respectively.

## 2 The Geostationary Interferometric Infrared Sounder (GIIRS)

### 2.1 GIIRS

The FY-4 satellites series are China's second-generation geostationary meteorological satellites with improved capabilities for weather and environmental monitoring. FY-4B, the second satellite in the FY-4 series was launched in June 2021, following FY-4A which was launched in December 2016. The GIIRS onboard FY-4 is an infrared Fourier transform spectrometer based

on a Michelson interferometer, also the first space-borne interferometer in geostationary orbit, primarily aiming to measure the three-dimensional atmospheric structure of temperature and water vapor for the numerical weather forecast. FY-4B/GIIRS is located at an altitude of 35,786 km above the equator at 123.5°E after launch and was relocated to 133°E after April 11, 2022. The observation domain of FY-4B/GIIRS is mostly over East Asia, with a focus on China, as shown in **Fig. 1(a).** Maps

of surface pressure and temperature, which show large range of variability in the observation domain, are presented in **Fig. A1** in the **Appendix**. FY-4B/GIIRS makes routine observations of the full region every 2 hours and 12 times per day (starting at 0, 2, 4, …, 22h UTC, respectively). Note that the starting hours have been changed to 1, 3, 5, …, 23h UTC after September 06, 2022. Each full region observation comprises 12 horizontal scans, and each scan sequence consists of 27 fields-of-regards (FORs) plus one deep space (DS) and one internal calibration target (ICT) measurement. The DS and ICT measurements are used as two known radiation sources to radiometrically calibrate the Earth-observing spectra. One full region coverage takes about 1.5 hours, and in the following 0.5 hours of each 2-hour observing cycle, the sounder is operated in the external calibration mode for instrument performance validation. The layout of the 2-dimension infrared plane array detector for each FOR is shown in **Fig. 1(b)**. The detector has 16×8 pixels with a sparse arrangement. A pixel spans 120 μm and the field of view (FOV) is 336 μrad. The spatial sampling on the Earth's surface is about 12 km at Nadir. The observed Earth's upwelling infrared radiation covers two spectra regions: long-wave IR band from 680 to 1130 cm$^{-1}$ and mid-wave IR band from 1650 to 2250 cm$^{-1}$ with a spectral resolution of 0.625 cm$^{-1}$. With low instrument noise and a high spectral resolution and range similar to current LEO IR sounders, GIIRS is in principle capable of measuring trace gases, including CO, and providing full day-night diurnal cycle observations. **Fig. 1(c)** shows an example of GIIRS spectra for the CO retrieval window from 2143 to 2181.25 cm$^{-1}$, and the Jacobian for CO and the interference gas $H_2O$ to demonstrate their contribution to the absorption features in the original spectra. In this micro-window, the absorption features from CO and the primary interference gas $H_2O$ are mostly separated and can be distinguished. Examples of Jacobian as a function of pressure and absorption strength are shown in the supplementary **Fig. S1**. The changing Jacobian values demonstrate the vertical sensitivity of the CO absorption lines at different pressure levels, which peak at the surface layer in the daytime and around mid-troposphere in the nighttime.

## 2.2 Assessment of GIIRS pre-launch instrument noise

For predicting FY-4B/GIIRS's post-launch performance, a series of blackbody calibration experiments have been conducted before launch in a laboratory thermal vacuum tank to evaluate the radiometric performances of the GIIRS instrument. As described in detail by **Li et al. (2022)**, the evaluation results showed that the noise equivalent differential radiance (NedR) on average in the mid-wave IR bands, covering the CO absorption channel, is less than 0.1mW/(m$^2$·sr·cm$^{-1}$). As for the radiometric calibration, the mid-wave IR band is susceptible to noise when the instrument is used to observe low-temperature targets. Nevertheless, the radiometric noise in brightness temperature also met the 0.7K requirement in the range of 260-315K, which is comparable to existing infrared sounders. As a result, low instrument noise for GIIRS makes it possible to provide strong constrain on retrieving CO vertical distribution.

## 2.3 Filtering of cloudy GIIRS pixels

Only clear-sky or near-clear-sky pixels are considered in the retrieval algorithm. To filter out cloudy pixels, we adopted the higher-resolution (4 km) level-2 cloud mask (CLM) data product from the Advanced Geostationary Radiation Imager (AGRI)

onboard FY-4B. AGRI uses multispectral threshold algorithms based on different spectral characteristics of VIS, NIR, and TIR bands under cloudy and clear conditions to obtain cloud mask information (**Lai et al., 2019**). The cloud mask of AGRI classifies pixels into four categories: clear, probably clear, probably cloudy, and cloudy. We collocated the GIIRS and AGRI footprints and assigned the GIIRS to be clear or near-clear when at least 80% of the collocated AGRI pixels are labeled as clear or probably clear. For each measurement cycle, there are 12×27 FORs and each FOR collects 16×8 observations using the infrared plane array detector. The total is 41472 observations. For each day with 12 measurement cycle, the total number of observations is about 500K. After cloud screening and excluding data with viewing zenith angle larger than 70°, the average daily number of clear sky observation is about 90K in July 2022.

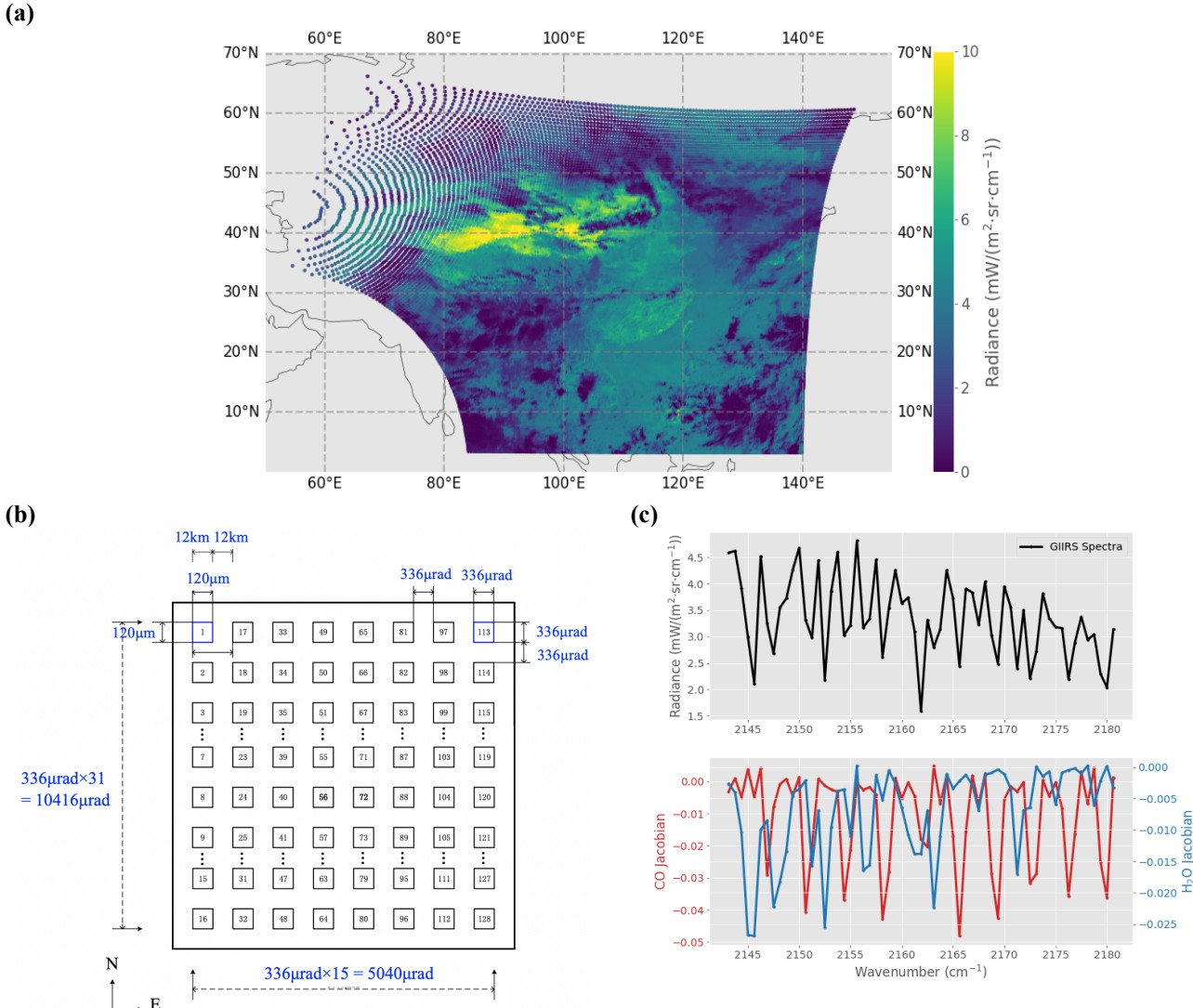

**(a)**

**(b)**

**(c)**

Figure 1: (a) FY-4B/GIIRS observation coverage from the tropics to ~60°N, and from ~60°E to ~140°E. The footprints on the left are sparse because of the geometric effect related to the large viewing zenith angle. The color denotes the observed radiance at 2132.5 cm$^{-1}$ in hour 04-05 UTC on July 24, 2022, as an example; (b) The layout of the 2-dimension infrared plane array detector of GIIRS. The detector has 16×8 pixels with a sparse arrangement. A pixel spans 120 μm and the field of view is 336 μrad. The spatial sampling on the Earth's surface is about 12 km at Nadir; (c) (top) An example of GIIRS spectra in the CO retrieval window from 2143 to 2181.25 cm$^{-1}$; (bottom) Jacobian at different channels for CO and the interference gas $H_2O$ in the 4$^{th}$ layer, where the averaging kernel value peaks.

## 3 The forward model in FY-GeoAIR for simulating observed spectra

An accurate radiative transfer (RT) model for simulating spectra is a prerequisite for constructing the inversion system for atmospheric composition retrieval. The thermal RT model is built based on radiative transfer theory with inputs of (1) atmospheric state including profiles of temperature, water vapor, atmospheric composition, and surface features; (2) the instrumental specifications, such as instrument spectral response function, and observing geometries; (3) spectroscopic database for computing the absorption cross-sections of gas molecules.

### 3.1 Radiative transfer in the thermal infrared

The upwelling spectral radiance observed by GIIRS can be computed by integrating the RT Equation (1), which includes the same processes as the forward RT models by TES (**Clough et al., 2006**) and IASI (**Hurtmans et al., 2012**). Under clear conditions, scattering by clouds and aerosols can be ignored. For the CO absorption window around 2150 cm$^{-1}$, the surface reflected solar radiation accounts for several percent in the total upwelling radiance and, therefore, cannot be neglected. The upwelling radiance received by a nadir-viewing satellite includes four main components: (a) surface emission (1$^{st}$ term in the r.h.s. of **Eq. (2)**); (b) upwelling atmospheric emission from the bottom- to the top- of the atmosphere [2$^{nd}$ term in the r.h.s. of **Eq. (1)**]; (c) surface-reflected downwelling atmospheric emission [2$^{nd}$ term in the r.h.s. of **Eq. (2)**]; and (d) surface-reflected solar radiation [3$^{rd}$ term in the r.h.s. of **Eq. (2)**]. All of these radiation sources are attenuated by the atmosphere. The RT Equation is given by:

$$I_v^\uparrow(\tau = 0, \mu) = I_v^\uparrow(\tau_v^*, \mu) \cdot T_v\left(\frac{\tau_v^*}{\mu}\right) + \int_0^{\tau_v^*} B_v(t(\tau')) \cdot \frac{\partial T_v\left(\frac{\tau'}{\mu}\right)}{\partial \tau'} d\tau' , \tag{1}$$

and $I_v^\uparrow(\tau_v^*, \mu)$ is the upwelling radiance from the surface layer comprising three sub-processes given by:

$$I_v^\uparrow(\tau_v^*, \mu) = \epsilon_v \cdot B_v(t_{skin}) + (1 - \epsilon_v) \cdot \tilde{I}_v^\downarrow(\tau_v^*) + \alpha_v \cdot I_v^{\downarrow\odot}(\tau_v^*) , \tag{2}$$

in which $\tau$ is the optical depth, and $\tau=0$ and $\tau_v^*$ represents the top and the bottom of the atmosphere, respectively; $\mu$ is the cosine of the satellite viewing zenith angle; $B_v(t)$ is the Planck function for computing black-body radiation at temperature t; $T_v(\tau)$ is the transmission at level $\tau$; $\epsilon_v$ is the emissivity; $\alpha_v$ is the surface reflectance; $I_v^{\downarrow\odot}(\tau_v^*)$ is the solar radiation reaching the surface level; $\tilde{I}_v^\downarrow(\tau_v^*)$ is the total downwelling flux reaching the surface, integrated upon all the geometries by considering a Lambertian surface. Similar to **Clough et al. (2006)** and **Hurtmans et al. (2012)**, the evaluation of this equivalent downward flux integral can be simplified by computing an effective downward radiance with a zenithal angle of 53.51°, which gives a very accurate approximation of the integral for emissivity larger than 0.9 (**Turner, 2004**).

The monochromatic radiances at a high resolution of 0.05 cm$^{-1}$ (over-sampled by 12 times compared to GIIRS spectral resolution of 0.625 cm$^{-1}$) are simulated and then convolved with the GIIRS instrument line shape (ILS) to obtain calculated radiances at the same resolution as GIIRS that can directly be compared with the GIIRS spectra. ILS for GIIRS is constructed using a standard SINC function with a Maximum Optical Path Difference (MOPD) of 0.8 cm. The original spectra are not

apodized to retain the original spectral absorption features. Instead, a wide ILS, with a width of 40 cm$^{-1}$, is used to account for the contribution from oscillating side lobes on both sides of the SINC function (**Gambacorta and Barnet, 2018**).

## 3.2 A priori atmospheric and surface parameters

The RT model in FY-GeoAIR requires inputs of atmospheric state parameters including profiles of temperature, water vapor, and atmospheric composition and surface parameters. The data sources and their specifications are described below.

### 3.2.1 The a priori CO profile and the associated covariance matrix

Since CO is the primary gas to be retrieved, constructing an appropriate a priori for the retrieval algorithm is important. There are currently two different approaches: (1) a fixed a priori CO profile for all retrievals. This approach has been used by IASI CO retrieval algorithm (**Hurtmans et al., 2012**). A static a priori is found to be more sensitive to unexpected events such as wildfires that lead to high CO emissions, because the variability associated with the a priori profile is larger in the retrieval algorithm (**George et al., 2015**); (2) spatially and temporally varying a priori CO fields from a model-derived monthly climatology. This approach has been used by the retrieval algorithm of TES (**Luo et al., 2007**) and MOPITT (**Deeter et al., 2010**); A climatology-based a priori provides the best a priori knowledge from the model and has been shown to have a better performance over regions with persisting high levels of CO throughout the year, such as urban areas (**George et al., 2015**).

However, time-varying a priori profiles make the retrieval results more complicated to interpret and to use for model validation, and the smaller variability associated with the a priori also makes it less sensitive to anomalies. Although A variable a prior profile can better capture variability and seasonality at different latitudes, it may not reflect the exact information existed in the observed spectra. For our purpose of retrieving the diurnal changes of CO columns, the main topic of this study, a fixed a priori is preferred because any significant perturbation to the constant a priori, which does not change diurnally, may indicate information that is retrieved from the observed spectra. To construct a fixed a priori profile and the associated covariance matrix, we used the CO simulations from the ECMWF Atmospheric Composition Reanalysis 4 (EAC4) monthly averaged fields (**Inness et al., 2019**), which have a horizontal resolution of 0.75°×0.75°, a temporal resolution of 3-hour, and 25 pressure levels from 1000 hPa to 1hPa. Data for the year 2021 over the land region are used, except the Tibet Plateau region where the CO is constantly low due to its high elevation. Over the ocean, we use the East China Sea only to avoid oversampling of ocean CO profiles. Noted that the EAC4 has assimilated MOPITT and IASI retrievals which can capture wildfire information. However, such information has almost completely reduced in the resulted a priori profile which is averaged from a large number of simulated profiles with the majority not affected by wildfire emissions. The mean and one standard deviation of the simulated CO profiles are used as the a priori and the associated error, respectively. This fixed a priori profile is used for different time and locations and no seasonality is assumed. To construct the correlation matrix, we used a correlation length of 3 km based on our analysis using EAC4 reanalysis. The covariance matrix can be calculated based on the a priori error and the correlation matrix. The resulted CO a priori and the associated covariance matrix are shown in

**Fig. 2**. For pressure levels that do not match the values shown in **Fig. 2(a)**, interpolation for pressure levels is carried out. Although the forward radiative transfer model (**Section 3.2.4**) has a full 47-layer atmosphere from 1000 hPa to 1hPa, we only retrieve the layers below 200 hPa and keep the layers above as the a priori.

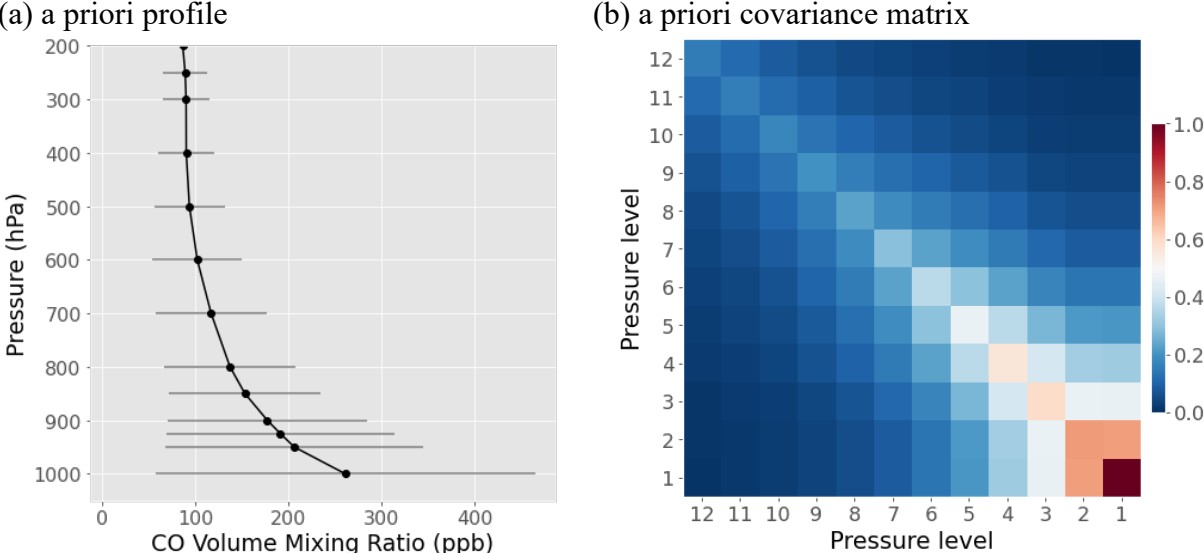

(a) a priori profile       (b) a priori covariance matrix

**Figure 2. (a) The a priori CO profile in volume mixing ratio from surface to about 200 hPa. The 12 levels corresponds to 11 layers and each layer has a thickness of about 1km; (b) the associated covariance matrix in unit less multiplicative factor for the 12 levels in (a). In the retrieval algorithm, only the layers below 200 hPa are retrieved while keeping the layers above as the a priori.**

### 3.2.2 Atmospheric profiles of $H_2O$, $CO_2$, $N_2O$, and $O_3$

In the spectral window from 2143 cm$^{-1}$ to 2181.25 cm$^{-1}$, the interference gases include $H_2O$, $CO_2$, $N_2O$, and $O_3$. $H_2O$ and $O_3$ are extracted from ECMWF ERA5 reanalysis (**Hersbach et al., 2020**), which has a horizontal resolution of 0.25°×0.25°, a temporal resolution of 1 hour, and 37 pressure levels from 1000hPa to 1hPa. $N_2O$ and $CO_2$ are extracted from ECMWF CAMS global inversion-optimized greenhouse gas fluxes and concentrations (**ECMWF, 2022**), which have a spatial resolution of 1.9°×3.75°, a temporal resolution is 3-hour, and 39 pressure levels from 1000hPa to 1hpa. The latest available year of datasets in 2019 for $N_2O$ and 2021 for $CO_2$ are used.

### 3.2.3 Atmospheric temperature profile

The atmospheric temperature profile is a key input in the forward RT model for computing the blackbody emission by the atmosphere. The atmospheric temperature data are extracted from ECMWF ERA5 reanalysis (**Hersbach et al., 2020**), which has a horizontal resolution of 0.25°×0.25°, a temporal resolution of 1 hour, and 37 pressure levels from 1000hPa to 1hPa.

### 3.2.4 Surface emissivity, surface skin temperature, and surface pressure

Surface emissivity and skin temperature are important parameters in computing surface blackbody emission. For the surface land data, we used the global infrared land surface emissivity database from the University of Wisconsin-Madison (UOW-M) (**Seemann et al., 2007**). The dataset has 10 bands in wavenumber, ranging from mid-wave infrared to long-wave infrared. Interpolation is performed to derive the emissivity in the CO channels. For ocean surface emissivity, we adopted the ocean emissivity model, which is a function of viewing zenith angle and wind speed, from **Masuda et al. (1997)**. Surface skin temperature and surface pressure are extracted from ERA5 hourly data on single level (**Hersbach et al., 2020**), which has a spatial resolution of $0.25° \times 0.25°$. The surface pressure from ECMWF reanalysis has a typical accuracy of 2–3hPa (**O'Dell et al., 2012**).

The number of pressure grids in the forward RT model should be large enough to reduce the error associated with the discretization of the atmosphere (**Clough et al., 2006**). Here we define a 47 layers target atmosphere with an equal thickness of about 1 km from 1000 hPa to 1hPa. All the above-described atmospheric a priori profiles are interpolated to the target pressure grids. The pressure grids are kept fixed except for the surface level which is determined by the surface pressure.

### 3.3 Spectroscopic database: Look-up tables of absorption cross-section

Deriving the absorption optical depth of gas molecules for the RT model would require the line-by-line calculation of absorption cross section based on spectroscopic line parameters and line shape. However, this line-by-line calculation at high spectral resolution for a wide spectral window is computationally expensive. Instead, to speed up the calculation, absorption coefficient (ABSCO) for different molecules at different pressures and temperatures are precalculated and stored in lookup tables (LUTs). For gas absorptions that have $H_2O$ dependence, the ABSCO dependence on $H_2O$ is also considered. This method of building ABSCO lookup tables has been adopted in previous retrieval algorithms, including the FORLI for IASI (**Hurtman et al., 2012**) and the ELANOR for TES (**Clough et al., 2016**).

In this study, ABSCO LUTs are built using the extensively validated Line-By-Line Radiative Transfer Model (LBLRTM v12.11; **Clough et al., 2005**). LBLRTM uses the HITRAN database as the basis for line parameters. These line parameters from HITRAN, plus additional line parameters from other sources, are combined for LBLRTM by a line file creation program called LNFL (v3.2). In addition to modeling individual spectral lines and absorption cross-sections, LBLRTM also takes into account the $H_2O$, $CO_2$, $O_2$, and $N_2$ continua in the thermal infrared using the MT_CKD continuum database (MT_CKD_3.4). The self-broadening absorption of $H_2O$ nonlinearly depends on its concentration which should be considered for calculating $H_2O$ ABSCO. It has been shown that the dependence is nearly linear for a given temperature and pressure. Therefore, to account for this effect in the LUT, we computed the $H_2O$ ABSCO at two $H_2O$ volume mixing ratio (VMR) values: 1ppm (dry air) and $4 \times 10^4$ ppm (wet air). The ABSCO values at other $H_2O$ values can be calculated by linear interpolation. In LBLRTM, the line-by-line calculation resolution is set to be $2.0 \times 10^{-4}$ cm$^{-1}$ and later integrated into the ABSCO LUT table resolution at $5.12 \times 10^{-2}$ cm$^{-1}$, which is oversampled by about 10 times compared to the GIIRS resolution of 0.625 cm$^{-1}$. The LUTs are built

for 49 atmospheric pressure levels from 1025 hPa to 1 hPa with a pressure step equivalent to 1 km, and 15 temperatures from 180 to 320 K with a step of 10K.

## 4 Retrieval algorithm in FY-GeoAIR based on optimal estimation theory

The goal of the retrieval algorithm for retrieving CO from nadir-viewing instruments based on optimal estimation theory is to find a solution for the state vector, which consists of CO profile and auxiliary parameters, such that the RT simulations best
fit the measured spectra. The optimal estimation method has been described thoroughly in **Rodgers (2000)** and applied in several previous studies by the group (**Zeng et al., 2017; Zeng et al., 2021; Natraj et al., 2022**).

### 4.1 Atmospheric inversion based on optimal estimation

The goal of optimal estimation is to find the solution for the state vector that minimizes the following cost function (**Rodgers, 2000**):

$$265 \quad J(\boldsymbol{x}) = \chi^2 = [\boldsymbol{y} - \mathbf{F}(\boldsymbol{x}, \boldsymbol{b})]^T \mathbf{S}_\varepsilon^{-1} [\boldsymbol{y} - \mathbf{F}(\boldsymbol{x}, \boldsymbol{b})] + (\boldsymbol{x} - \boldsymbol{x}_a)^T \mathbf{S}_a^{-1} (\boldsymbol{x} - \boldsymbol{x}_a), \tag{3}$$

where $\boldsymbol{y}$ is the observed GIIRS spectral radiance at the CO retrieval window from 2143 cm$^{-1}$ to 2181.25 cm$^{-1}$, which is found to be the best window that minimizes interferences by other gases while maximizing the information content for CO retrieval (**De Wachter et al., 2012**); $\boldsymbol{x}$ is the state vector consisting a set of parameters to be retrieved, including CO profile, $H_2O$ profile, surface skin temperature and atmospheric temperature profiles. Of the 47 layers atmosphere defined in the forward model, the
270 algorithm only retrieves layers below 200 hPa (in total of 11 layers at maximum) and uses the a priori for the layers above. $\mathbf{F}$ is the forward RT model for simulating radiance as introduced in **Sect. 3**; $\boldsymbol{b}$ is a set of model parameters in the RT model that are not retrieved, such as profiles of interference gases ($O_3$, $CO_2$, and $N_2O$), surface emissivity, and other relevant geophysical parameters. $\varepsilon$ is the spectral error vector containing the noise in the spectra observation. $\mathbf{S}_\varepsilon$ is the measurement error covariance matrix; $\boldsymbol{x}_a$ is the *a priori* state vector; $\mathbf{S}_a$ is the *a priori* covariance matrix for the state vector. For simplicity in calculating $\mathbf{S}_\varepsilon$,
we assume that the measurement noise dominates and there is no cross-correlation between different spectral channels, resulting in a diagonal matrix. The instrument noise (NedR) for each spectral observation, as described in **Sect. 2.2**, is used as the measurement noise. A commonly used statistic for quantifying the goodness of fit is the reduced $\chi^2$, which is computed as the cost function value after convergence divided by the degree of freedom, which is the number of channels in the absorption window (which is 64) minus the number of elements in the state vector (which is 4). After evaluating the reduced $\chi^2$ from test
runs, we found that the value on average is systematically lower than the theoretical mean value of 1.0, indicating that the measurement error may have been underestimated. Therefore, in this study, we enlarge the measurement noise by 1.5 times such that the averaged reduced $\chi^2$ value from the retrievals is close to 1.0. The extra noise added may represent the uncertainty from the forward model, spectroscopy, and the forward model inputs, which are not accounted for by the original instrument

noise alone. $\mathbf{S}_a$ is a very important parameter that should not be too tight or loose to provide suitable constrain on the retrieval. It can be calculated based on the error in CO a priori profile and the correlation matrix derived from model simulations, as described in **Section 3.2.1**. To obtain the solution for **Equation (3)**, we adopt the Levenberg-Marquardt method (**Rodgers, 2000**) to find the optimal estimate of $x$.

## 4.2 Averaging kernel (AK) matrix and degree of freedom for signal (DOFS)

The quality of the retrieval can be characterized by two quantities: the AK matrix and the DOFS. AK matrix is an important statistical metric for describing the sensitivity of the retrieval to the true state by the current observing system. The full averaging kernel matrix ($m \times m$) is given by:

$$\mathbf{A} = (\mathbf{K}^T \mathbf{S}_\varepsilon^{-1} \mathbf{K} + \mathbf{S}_a^{-1})^{-1} \mathbf{K}^T \mathbf{S}_\varepsilon^{-1} \mathbf{K}, \tag{4}$$

where $m$ is the number of atmospheric layers. $\mathbf{K}$ is the Jacobian matrix, which is the first derivative of the forward model with respect to the state vector. $A_{ij}$ represents the derivative of the retrieved CO at level $i$ with respect to the true CO at level $j$, representing a relative contribution of the true state to the retrieved state. An ideal observing system would produce an AK matrix close to the identity matrix, meaning the observations are sufficiently good to constrain each element in the retrieval vector. In reality, the AK can be very different from an identity matrix, meaning that the information from the true state is smoothed vertically over different layers by the retrieval algorithm. The rows of AK can be regarded as smoothing functions. The trace of the AK matrix, representing the number of independent elements of information extracted by the retrieval algorithm from the measurement, quantifies the DOFS. It is an important concept in describing the vertical resolution of the retrieval profile. For example, a DOFS of 1 means that at least one independent piece of information on the vertical distribution of CO can be retrieved from the spectral measurement.

## 4.3 Post-processing

All cloud-screened GIIRS spectra acquired over land and ocean at solar zenith angle less than 70° are used in the retrieval. In the post-processing, multiple filters are applied to ensure good retrieval quality. First, retrievals that fail to converge after 10 iterations are excluded. Second, retrievals with the goodness of fit, quantified by reduced $\chi^2$, less than 1.5 are excluded. Lastly, retrievals with root-mean-square-error of the fitting BT residual that are more than one standard deviation away from the mean, which is about 0.7K in July 2022, are excluded. After data screening, the total number of observations in July 2022 (in total 2,812,071) is reduced to 2,045,228.

## 5 Inversion experiments using simulated synthetic spectra

The goal of applying the FY-GeoAIR algorithm to simulated synthetic spectra is to assess the performance of the algorithm in retrieving CO profiles and to quantify the impacts on the accuracy due to the spatially and temporally varying thermal contrast (TC) in East Asia. TC is defined as the temperature difference between the surface and the lower atmospheric layer (**Clarisse et al., 2010**), and is found to be a key indicator of the information content for retrieving CO profile, especially the lower tropospheric CO, from infrared thermal radiance.

Four representative regions are specially selected for inter-comparisons, including (1) North China Plain (covering 32°-40°N and 114°-120°E), which represents industrialized urban regions with persistently high CO emissions in China; (2) Mongolia (covering 42°-50°N and 100°-115°E), which represents CO background regions; (3) the East China Sea (covering 25°-33°N and 122°-129°E), which represents ocean surface; and (4) North India (23-28°N and 75°-83°E), which represents industrialized urban regions in India. The locations of these regions are shown in the Appendix **Figure A1(a)**. The diurnal changes of surface temperatures and bottom air temperatures, as shown in **Fig. 3(a)**, show distinctive patterns over these selected regions. Compared to the atmosphere, Earth's surface, especially over land, heats up and cools down more quickly because of its relatively low heat capacity. This mechanism results in a larger diurnal variation for the surface than for the atmosphere: TC is thus more pronounced during the day than the night. Specifically, the East China Sea has a relatively flat change, as expected for ocean water due to its large heat capacity and ocean water mixing; The Mongolia region, covered by a mixture of grass and bare land, has the largest diurnal change; The North China Plain and North India, surrounded by urban clusters with a mixture of residential and agriculture lands, has a moderate diurnal change. The complexity of the diurnal TC change as demonstrated by various land use types in East Asia affects the diurnal changes of DOFS from the CO retrievals by FY-4B/GIIRS, as shown in **Figure 3(b)**. TC is significantly correlated with the DOFS from the CO retrievals for both the total column and the lower atmospheric partial column (the bottom 3-layer in this case). A higher TC in the daytime results in a larger DOFS. The information content for the lower atmosphere shows a similar pattern to the total column but a larger relative change between daytime and nighttime. A similar relationship can be found in LEO satellites (e.g., **Bauduin et al., 2017**).

The synthetic spectra are generated using the same forward RT model as described in **Sect. 3**, except that we use the original ECMWF EAC4 3-hourly simulated CO as the "truth" and add Gaussian white noise with mean of zero and a standard deviation equal to NedR×1.5. Since we assume no error in the forward RT model, the spectra error solely comes from the added noise. The retrieval algorithm is then applied to these synthetic spectra using the fixed a priori CO as described in **Sect. 3.2**. The retrievals are compared with the "truth" to investigate the impacts of TC and AK on their differences.

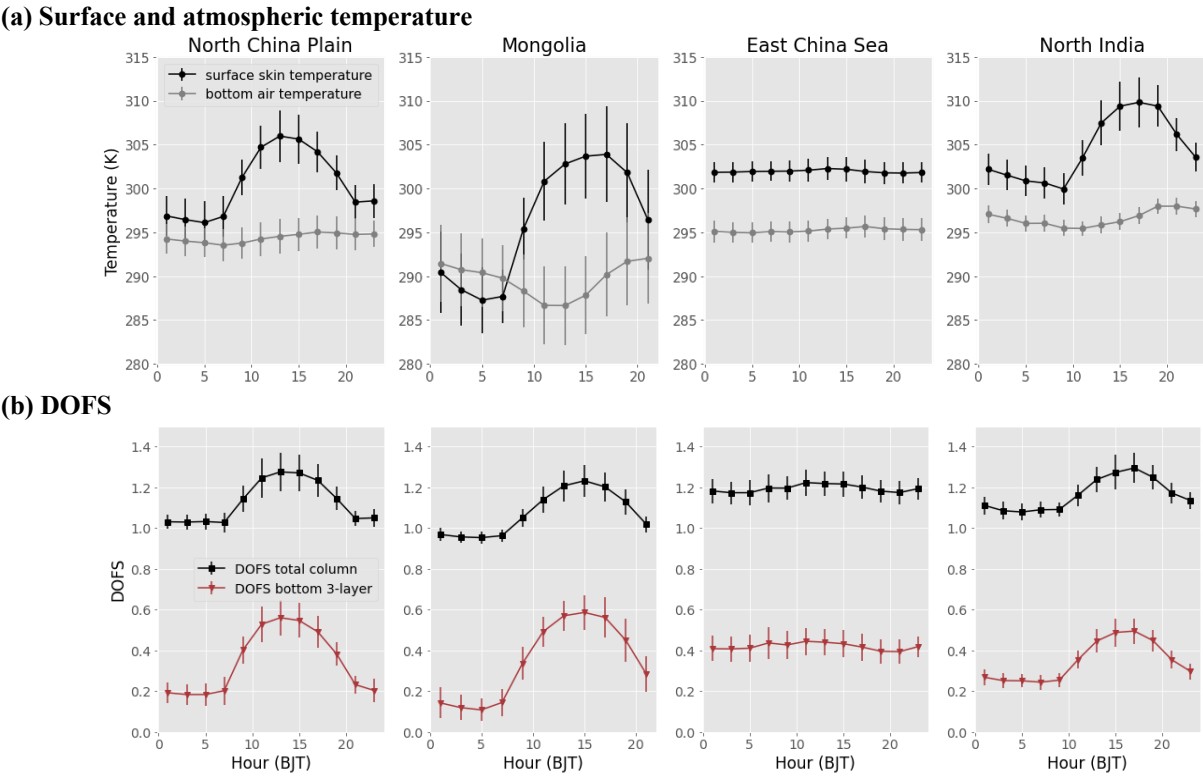

**(a) Surface and atmospheric temperature**

**(b) DOFS**

**Figure 3: (a) The diurnal change of surface temperature and bottom 2-layer mean air temperature extracted from ECMWF ERA5 reanalysis data for the four representative regions: North China Plain, Mongolia, East China Sea, and North India. These temperature values are averaged for every two-hour corresponding to the clear-sky GIIRS observations in July 2022. (b) The diurnal change of DOFS for the total column and the bottom 3-layer (from the surface to 3km a.s.l.) atmosphere. The error bars represent regional data variabilities. BJT represents Beijing Time (UTC+8).**

To evaluate the performance of the retrieval algorithm, we compare (1) the retrievals and the "truth" and (2) the a priori and the "truth". The comparisons in North China Plain are shown in Fig. 4, while the comparisons in Mongolia, East China Sea, and North India are shown in **Fig. S2**, **Fig. S3**, and **Fig. S4**, respectively. From **Fig. 4**, we can see that the retrieved CO total columns have a higher correlation with the "truth" than with the a priori column. Noted that the small variation of the a priori column derived from the fixed a priori profile is caused by the surface pressure difference within the region. The comparison suggests that the retrieval algorithm is effective and the observed spectra are providing useful information to constrain the CO profiles. In the daytime, when the DOFS is higher, the retrieved CO columns show the largest correlation and the smallest bias when compared with the "truth" columns. In the nighttime, however, DOFS becomes smaller, especially in the lower atmosphere. This low DOFS means that the CO profile retrievals are under-constrained and show a larger bias when compared with the "truth" columns. The information for the bottom layer CO may be extrapolated from the free troposphere (**Bauduin et al., 2017; George et al., 2015**). This information extrapolation may lead to a higher bias in the retrieval for the bottom layer

CO. Overall, we can conclude that a higher (lower) TC leads to a higher (lower) DOFS for the retrieval, which in turn contributes to a smaller (larger) bias in the retrieval.

Since the sensitivity of the CO retrieval is pressure dependent, we implement a profile correction by applying the GIIRS AK to smooth the "truth" profile, to account for the different resolution between the retrieval and the "truth" (**Rodgers and Connor, 2003**):

$$CO_{true}^{AKcorrect} = CO_a + \mathbf{A} \cdot (CO_{true} - CO_a) \, , \tag{5}$$

where $x_{true}^{AKcorrect}$ is the smoothed "truth" profile and $x_a$ and $\mathbf{A}$ are the a priori profile and AK matrix, respectively, from the GIIRS retrieval. The result is shown in the 3rd column in **Fig. 4**. The correlations are significantly improved, justifying the use of AK matrix smoothing for the CO profile from model simulations that have uniform vertical sensitivity. For example, if the model simulation data are close to the "truth", then after the AK matrix smoothing, the corrected data should be in high agreement with our GIIRS retrievals.

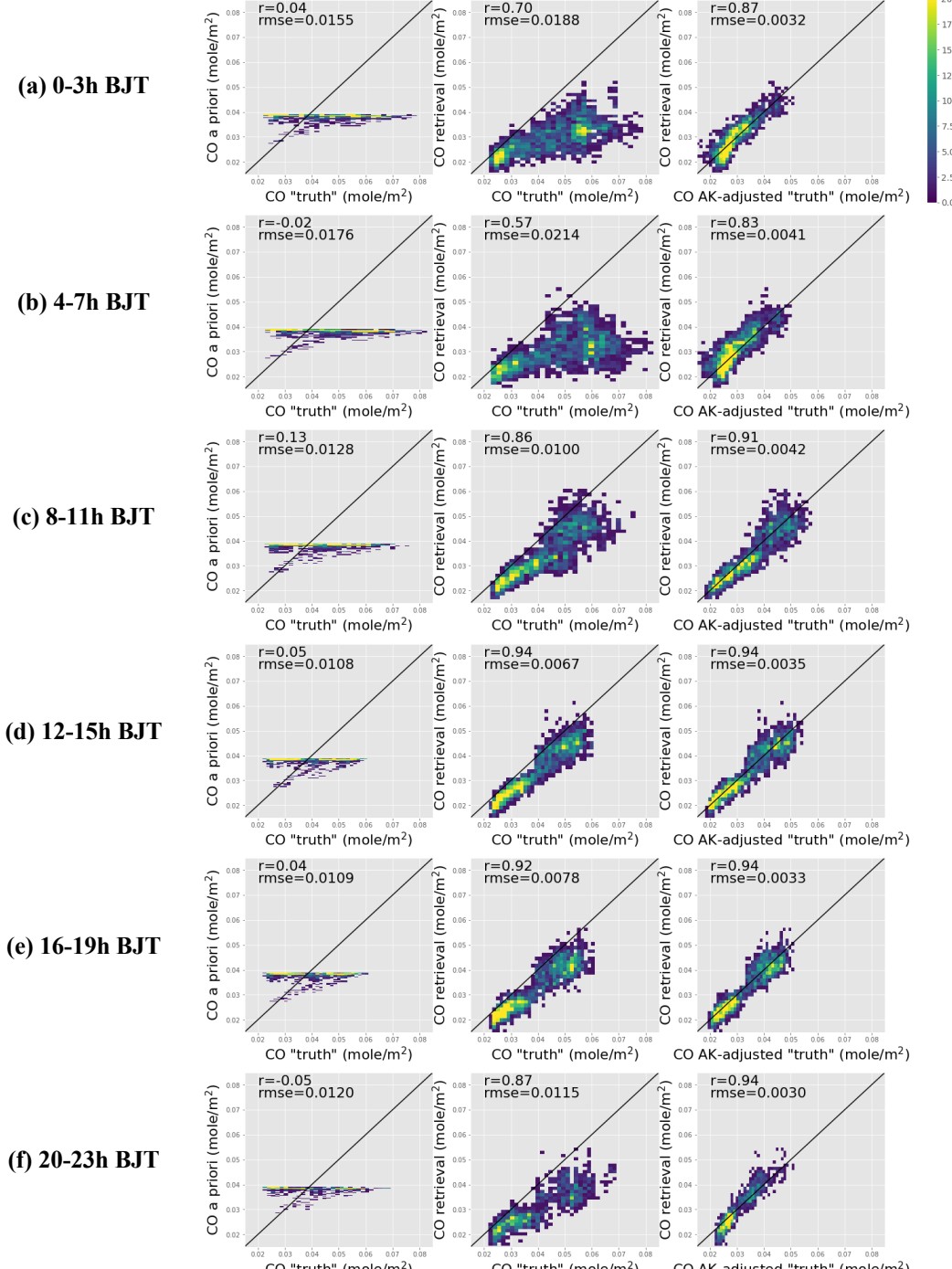

**Figure 4. Results from inversion experiments in North China Plain. (left) Comparison between CO a priori total column and CO column "truth" ; (middle) comparison between CO total column retrieval and CO column "truth", and (right) comparison bewteen CO total column retrieval and the "truth" after AK-smoothing. These results are from synthetic simulation experiments using data on July 7 and 24 of 2022. The results are shown for every 4 hours. BJT represents Beijing Time (UTC+8). Results from inversion experiments for Mongolia, East China Sea, and North India are shown in the supplementary materials.**

## 6. Characteristics of CO retrievals from GIIRS

All cloud-screened GIIRS spectra acquired over land and ocean with viewing zenith angle less than 70° are used in the FY-GeoAIR retrieval algorithm. The retrieval results have been post-screened by the filters described in **Sect. 4.3**. In this section, we describe the characteristics of the CO retrievals, including the goodness of spectral fitting, DOFS and vertical sensitivity using AK matrix, and accuracy assessment by comparison with IASI CO retrievals.

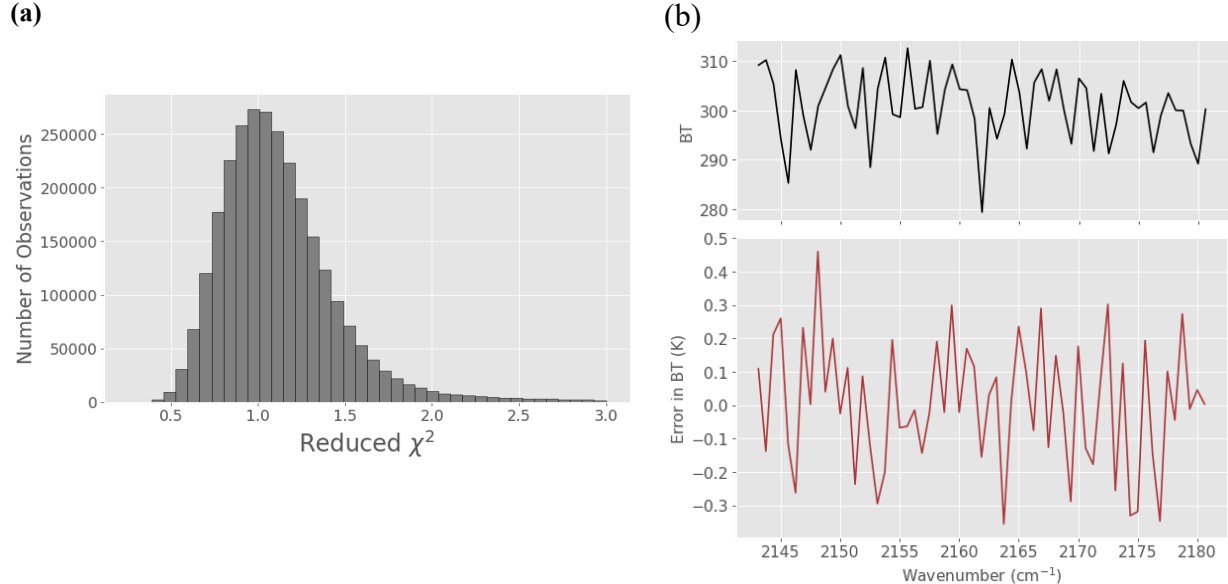

**(a)**  **(b)**

**Figure 5: Statistics from spectral fitting. (a) The histogram of reduced $\chi^2$ for spectral fitting from all retrievals; (b) The upper panel shows an example of the observed spectra in brightness temperature. The lower panel shows the spectral fitting residual in brightness temperature averaged over all post-screened retrievals in July 2022. The standard deviations of the fitting errors are consistent across different channels, which is about 0.6 K.**

### 6.1 Statistics for spectral fitting

The goodness of spectral fitting is evaluated by two statistics: the spectral fitting residual and the reduced $\chi^2$. The latter measures how large is the spectral fitting residual compared to spectral noise. The histogram of the reduced $\chi^2$ and the averaged fitting residuals for all the retrievals in July 2022 are shown in **Fig. 5**. The reduced $\chi^2$ shows an expected distribution centering around 1.0. In the post-processing step, we filter the values that are larger than 1.5. The choice of the threshold may vary, here we chose 1.5 to retain more good quality data while removing retrievals that have an unsatisfactory fitting. The spectral fitting errors in brightness temperature (BT) show no significant bias. Although a systematic pattern, that is persistent among observations at different hours, can be seen from the averaged fitting residual from all spectra. However, this pattern is not correlated with the absorption feature of the target gas CO or the primary interference gas $H_2O$ (**Fig. 1(c)**). This suggests

that the CO spectroscopy adopted from LBLRTM which has undergone extensive verifications is accurate for the purpose of this study. The standard deviations of fitting errors are very consistent across different channels, which is about 0.6 K for the BT error. This suggests that the majority of the channels have fitting errors of less than 1K. This result is consistent with the pre-launch assessment as described in **Sect. 2.2** and **Li et al. (2022)**.

## 6.2 Information content analysis

The information content of FY-4B/GIIRS retrievals is assessed by using the AK matrix and DOFS. The AK matrix represents the sensitivity of the retrieved state to the true state, in which the matrix row represents how a specific retrieved state vector element reacts to true changes of the state vector at different layers. In the case of the CO profile, for each retrieved layer, the AK peak at the altitude containing most information about the profile. The AK thus provides an estimation of the altitude of maximum sensitivity. **Fig. 6** shows examples of AK rows from two different scenarios of TC: higher TC in the daytime (TC=8.4K) and low TC in the nighttime (TC=1.0K). The distinctive difference in the lower tropospheric AK values demonstrates the importance of high TC in providing information to the lower tropospheric CO retrievals. For the low TC scenario, little information is available from the lower troposphere and the altitude with maximum sensitivity is located around 500 hPa (~5 km a.s.l.). In such a case, the information for CO in the lower troposphere is thus extrapolated from the mid-troposphere, which may lead to high bias for the lower troposphere estimate. **Fig. 7** shows that the diurnal (every 2-hour) change of AK diagonal elements in North China Plain, Mongolia, East China Sea, and North India averaged over all days in July of 2022. The diurnal changes are primarily driven by the changes in TC as shown in **Fig. 8**. As the TC increases and peaks in the afternoon, the AK diagonal value increases, and the layer with the maximum moves closer to the surface. In contrast, the East China Sea region has little change among different hours because the TC is relatively flat throughout the day. These results demonstrate that the TC is significantly correlated with the vertical structure of AK rows from FY-4B/GIIRS retrievals.

DOFS represents another important metric for information content from FY-4B/GIIRS spectra. It quantifies the number of independent information available from the measurement. **Fig. 8** shows the diurnal changes of the DOFS mean and standard deviation in the four representative regions averaged over all days in July of 2022. In general, the diurnal changes of DOFS track the TC changes over these regions (**Fig. 2**). Interestingly, we see that the North China Plain and North India have comparable or higher DOFS to Mongolia, although the latter has a significantly larger TC. This is because, besides TC, the high CO concentration in North China Plain and North India have a positive contribution to the DOFS, as also shown by **Bauduin et al. (2017)**.

To investigate the relationship between TC and DOFS, the DOFS from the retrievals as a function of TC for the CO total column and bottom 3-layer CO partial column for the four representative regions are plotted in **Fig. 9**. The DOFS for the majority shows a monotonic change with TC. However, for retrievals with negative TCs, the DOFS may increase as the TC becomes more negative. This pattern is consistent with findings by **Bauduin et al. (2017)**, which showed that large negative TC values allow the decorrelation between the low and the high troposphere by capturing the emission of radiation from the

lower troposphere. Overall, the DOFS for the total CO column retrieval is between 0.8 and 1.5 for the majority, with a mean value around 1.1, meaning that more than one independent piece of information is retrieved from FY-4B/GIIRS spectra. For

the bottom 3-layer ranging from the surface to 3km a.s.l., the DOFS for the majority is between 0 and 0.8. The highest DOFS exists over the land region with the largest TC. Similarly, for the DOFS for the bottom 3-layer, increasing the TC value favors the sensitivity to surface CO.

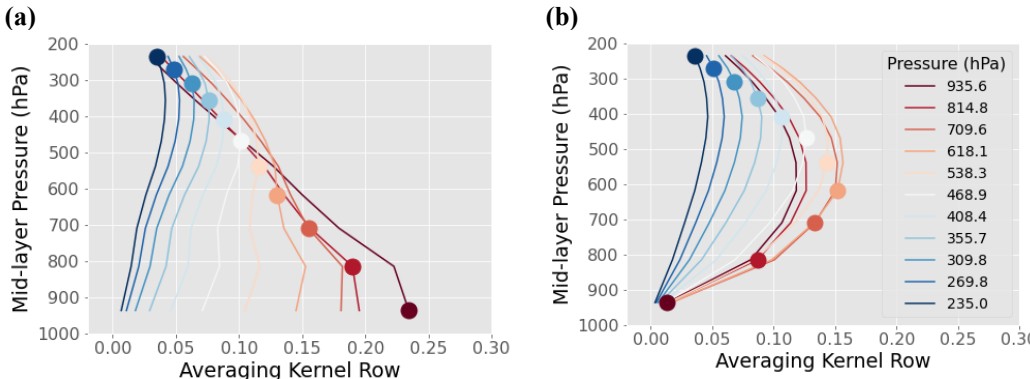

**Figure 6: Examples of averaging kernel rows from two different scenarios of TC: (left) high thermal contrast in the daytime (TC=8.4K) and (right) low thermal contrast around Beijing in the nighttime (TC=1.0K). These are the same two data points in the above figure (in the supplementary materials).**

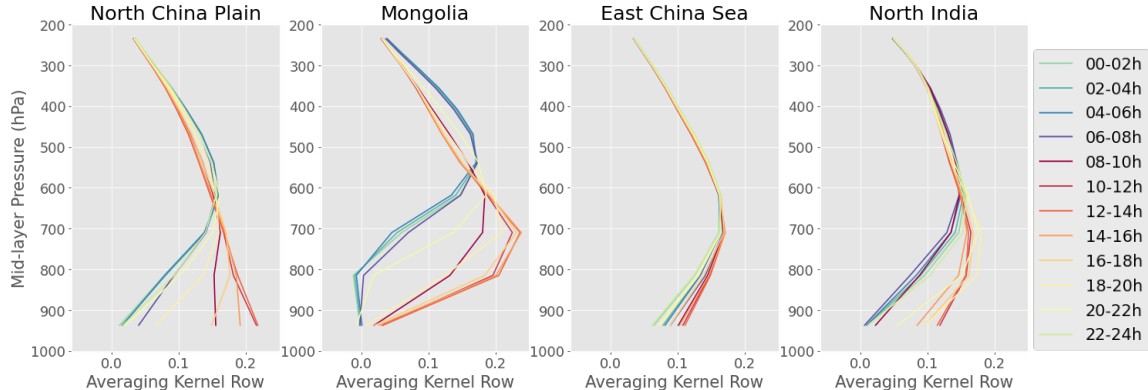

**Figure 7: Diurnal (every 2-hour) change of averaging kernel diagonal elements in North China Plain, Mongolia, East China sea, and North India averaged over all days in July of 2022. The hour information is based on Beijing Time (UTC+8).**

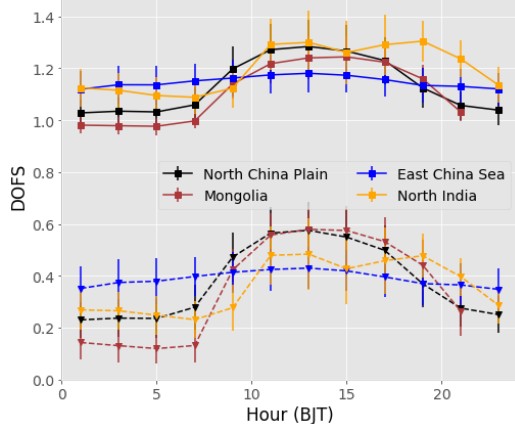

**Figure 8. The diurnal changes (every 2-hour) of the DOFS mean and standard deviations for (solid) CO total column and (dashed) the bottom 3-layer (from the surface to 3km a.s.l.) partial column in the four representative regions: North China Plain, Mongolia, East China Sea, and North India. These data are averaged over all days in July 2022.**

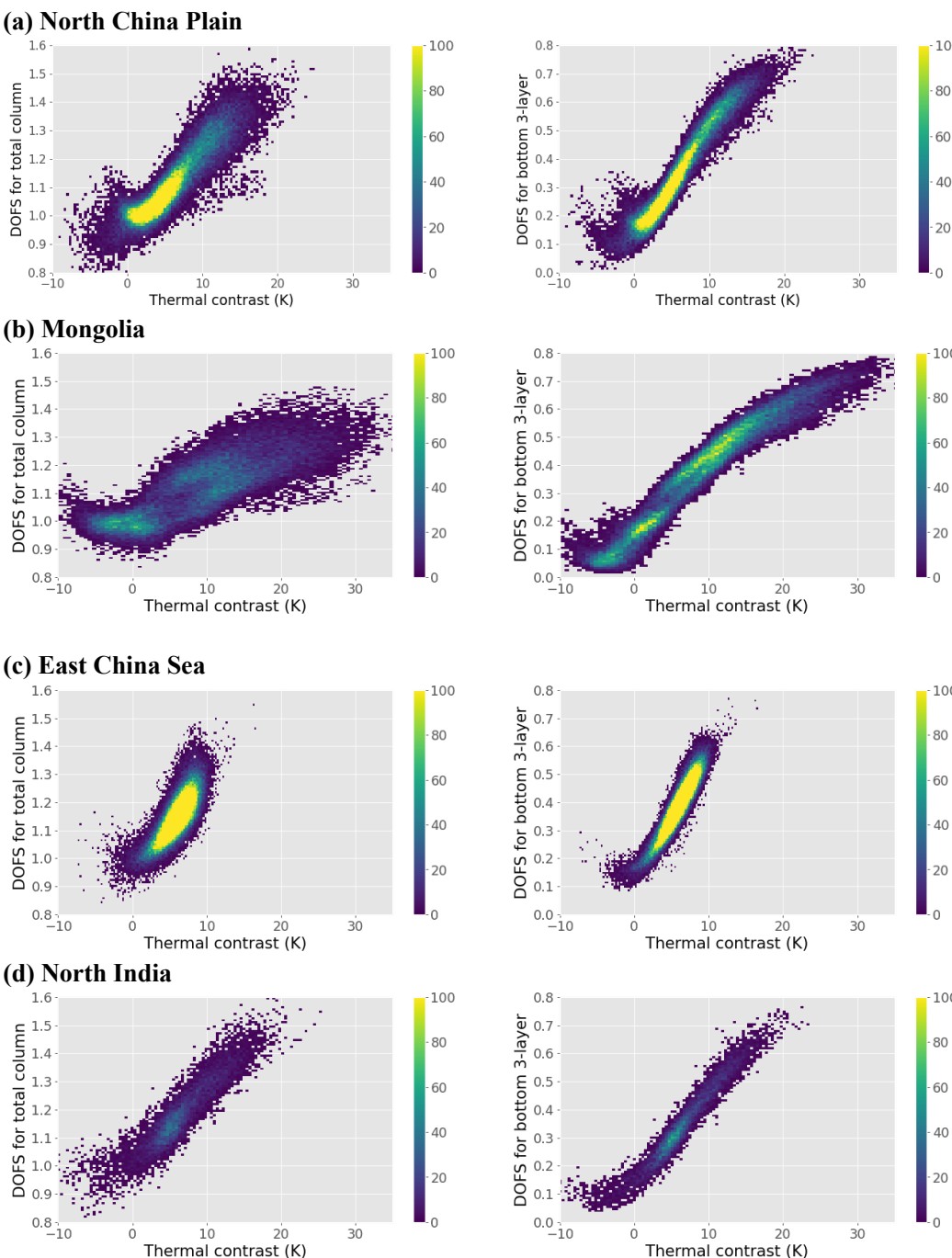

**(a) North China Plain**

**(b) Mongolia**

**(c) East China Sea**

**(d) North India**

**Figure 9: DOFS from the retrievals as a function of TC for CO total column (left) and bottom 3-layer (from the surface to 3km a.s.l.) CO partial column (right) for the three representative regions: (a) North China Plain, (b) Mongolia, and (c) the East China Sea, and (d) North India.**

## 6.3 Temporal and spatial variations of CO total column and the DOFS from retrieval

The diurnal changes of CO columns from FY-4B/GIIRS retrievals may be linked to (a) the real CO column variabilities, the objective of this study, primarily driven by emissions, atmospheric transport, and changes in boundary layer height, and (b) the change in instrument detectivity as quantified by the retrieval DOFS that is affected by TC and CO abundance. As an attempt to disentangle these different factors, in **Fig. 10** we show the diurnal changes of the CO column retrievals from FY-4B/GIIRS, the boundary layer height (BLH) from ECMWF reanalysis data, and the model CO simulations from EAC4 reanalysis over the selected regions. From the model simulations, which has assimilated satellite observations of IASI and MOPPIT, we see the change in total columns in all selected regions are very small (less than 2% on average) which can be primarily attributed to the diurnal change in BLH. In the supplementary **Fig. S6**, the model simulated ground-level CO concentrations show a much larger variation compared to the CO columns. However, model simulations from EAC4 have large grids ($0.75° \times 0.75°$) and low temporal resolution (3-hour) which are not sufficient to resolve the local CO changes that are usually measured over the urban centrals. To the first order, the change in the ground-level CO concentration averaged over the selected region is primarily driven by the BLH change, and the traffic emission peaks are not discernable from the time series.

In urban regions, ground level CO concentrations from in-situ ground-based observations show a distinctive double-peak diurnal cycle corresponding to the morning and evening traffic rush hours. In the morning, the increase in traffic emissions results in the morning peak; As BLH gets larger, air dilution takes place and the concentration drops; In the evening, traffic emissions increases while the BLH quickly decreases which results in an evening peak. These diurnal pattens have been observed cross many cities in Asia (e.g., **Ran et al., 2009; Chen et al., 2020; Meng et al., 2009; Verma et al., 2017**). However, the diurnal changes of CO columns and the surface layer concentration are not expected to be the same. As shown in **Stremme et al. (2009)**, which retrieved diurnal CO column changes in the Mexico City using ground-based solar and lunar infrared spectroscopy, found that the diurnal changes in the total CO column and the surface level concentration can be very different. The total CO column within the city presents large variations with contributions from urban CO emissions at the surface and the transport of cleaner or more polluted air masses into the study area.

The diurnal cycle of the retrieved CO columns from FY-4B/GIIRS, shown in **Fig. 10(a)**, presents impacts by the diurnal change of DOFS. Noted that the a priori CO columns averaged for the year of 2021 are 0.038, 0.028, 0.039, 0.036 mole/m$^2$, respectively, for North China Plain, Mongolia, East China Sea, and North India. Since the DOFS for the nighttime is low, especially for the lower atmosphere, the nighttime column retrievals generally tend to close to the a priori columns, resulting in a bow shape. In the daytime, the CO column retrievals are well constrained by the observations. Since the summer time CO is generally lower than the yearly mean (**Chen et al., 2020**) based on CO's seasonal cycle, we see the column retrievals averaged in July are generally lower than the a priori column value which is derived from annual mean of model simulations. These results suggest that a direct interpretation of the authentic diurnal column variabilities from the retrieved CO columns is challenging given the entangled effects of the real CO changes and the variable detectivity. A better solution is to use mode

assimilation (e.g., EAC4) that takes into account the retrieved CO profile and the vertical sensitivity in order to disentangle the different contributions.

We further compare the diurnal changes of the spatial distribution of (1) the column retrievals and (2) the DOFS values for both the total column and the bottom 3-layer partial column. The CO total column and the DOFS from the retrieval algorithm are averaged for every measurement cycle (2-hour duration) by aggregating the data from all days in July 2022 into $0.5° \times 0.5°$ grids. Noted that the GIIRS completes a measurement cycle in about 2 hours, so the data at a certain location is just an instantaneous value within the measurement cycle. Since CO is a relatively long-lived gas, we can assume the CO values are

unchanged during the 2-hour period. In total, 12 full domain measurement cycles in East Asia are available for every 2 hours. Here, we only show every other measurement cycle as examples, in total 6 full domain measurements for hours 0-2h, 4-6h, 8-10h, 12-14h, 16-18h, and 20-22h UTC. The results are shown in **Fig. 11(a)** and **Fig. 12(a)** for the CO total column retrievals. The CO columns have small changes in a day and the spatial pattern is very persistent. Overall, the spatial distribution of the CO total column shows expected spatial patterns, with high values clustered in industrialized urban regions in northern and

eastern China, the Sichuan basin in central China, and northern India. Over the ocean, the enhancement caused by the East Asian outflow (**Heald et al., 2003**) can be observed. In addition, elevated CO column values can be detected around the Siberia region which is close to the north eastern region of our study area. The high CO values are related to the wildfire emissions over the region which intensifies in the summer season.

    To evaluate the information content that is available in constraining the CO total column and the bottom 3-layer partial

column, the maps of the DOFSs are also shown in **Fig. 11(b)(c)** and **Fig. 12(b)(c)**. In the daytime, when TC is high, the DOFS values over land, especially over easter and northern China, are higher than 1.0, suggesting that the GIIRS spectra provide more than 1 piece of independent information to the retrieval. In the nighttime, when the land region cools down quickly and the ocean surface is still warm, a larger TC over the ocean leads to a higher DOFS compared to the land region. A similar pattern can be observed in the DOFS for the bottom 3-layer, but with smaller DOFS values. Not surprisingly, the DOFS for

the bottom 3-layer also shows a local pattern as the CO emissions (e.g., in northern China). This is because the DOFS from the retrieval is also affected by the CO concentration besides TC, as discussed in **Bauduin et al. (2017)**.

**(a) CO column**

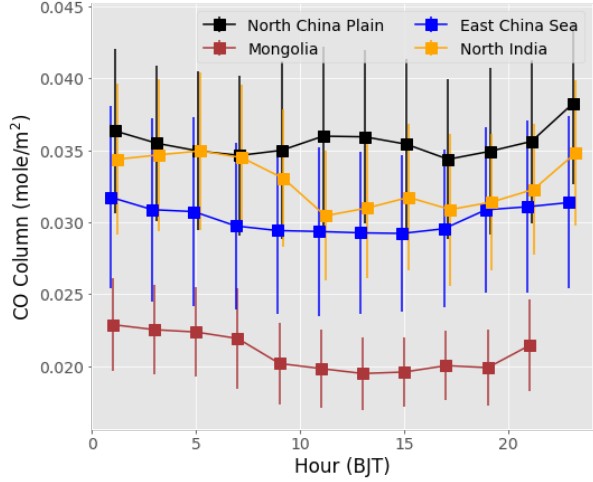

**(b) Boundary layer height**          **(c) Model CO simulations**

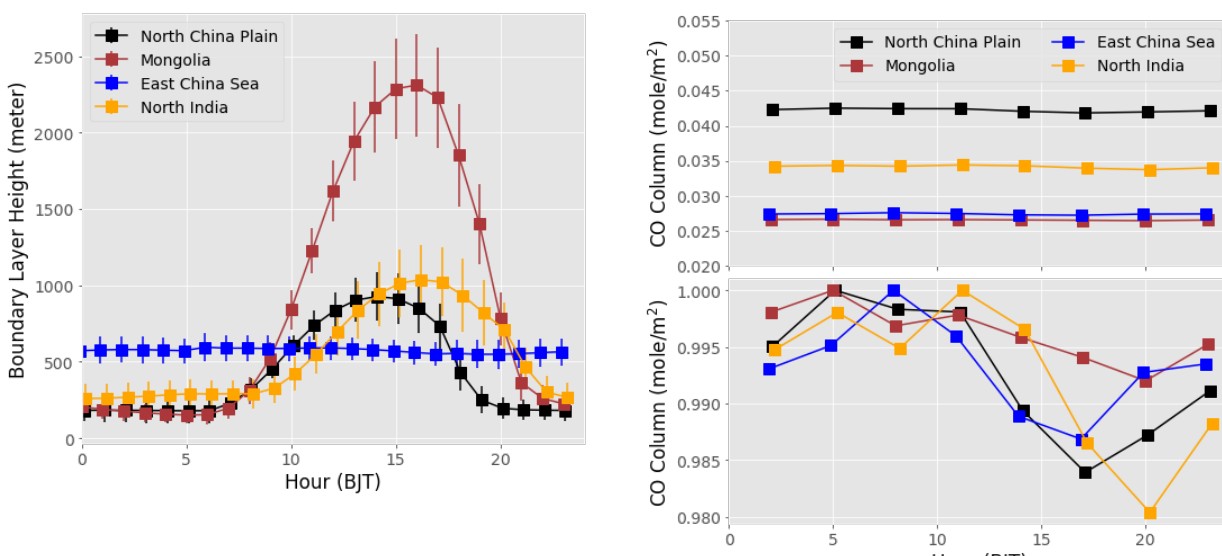

**Figure 10. (a) The diurnal changes (every 2-hour) of the retrieved CO columns averaged over all days in July 2022. The x-axis values are slightly shifted intentionally to avoid the overlapping of the error bars. Noted that no data are available in hour 23 for the Mongolia region; (b) The diurnal changes (every hour) of averaged boundary layer height in the representative regions in July 2022. The data are averaged from the ECMWF ERA5 boundary layer height reanalysis data; (c) The diurnal changes (every 3-hour) of simulated CO columns in the upper panel and the normalized values in the lower panel. The simulation data are averaged from the ECMWF EAC4 reanalysis dataset in July 2021. The recent 2022 data are not available at this point. The error bars in (a) and (b) represent day to day variabilities. The error bars for (c) are too large and therefore not shown.**

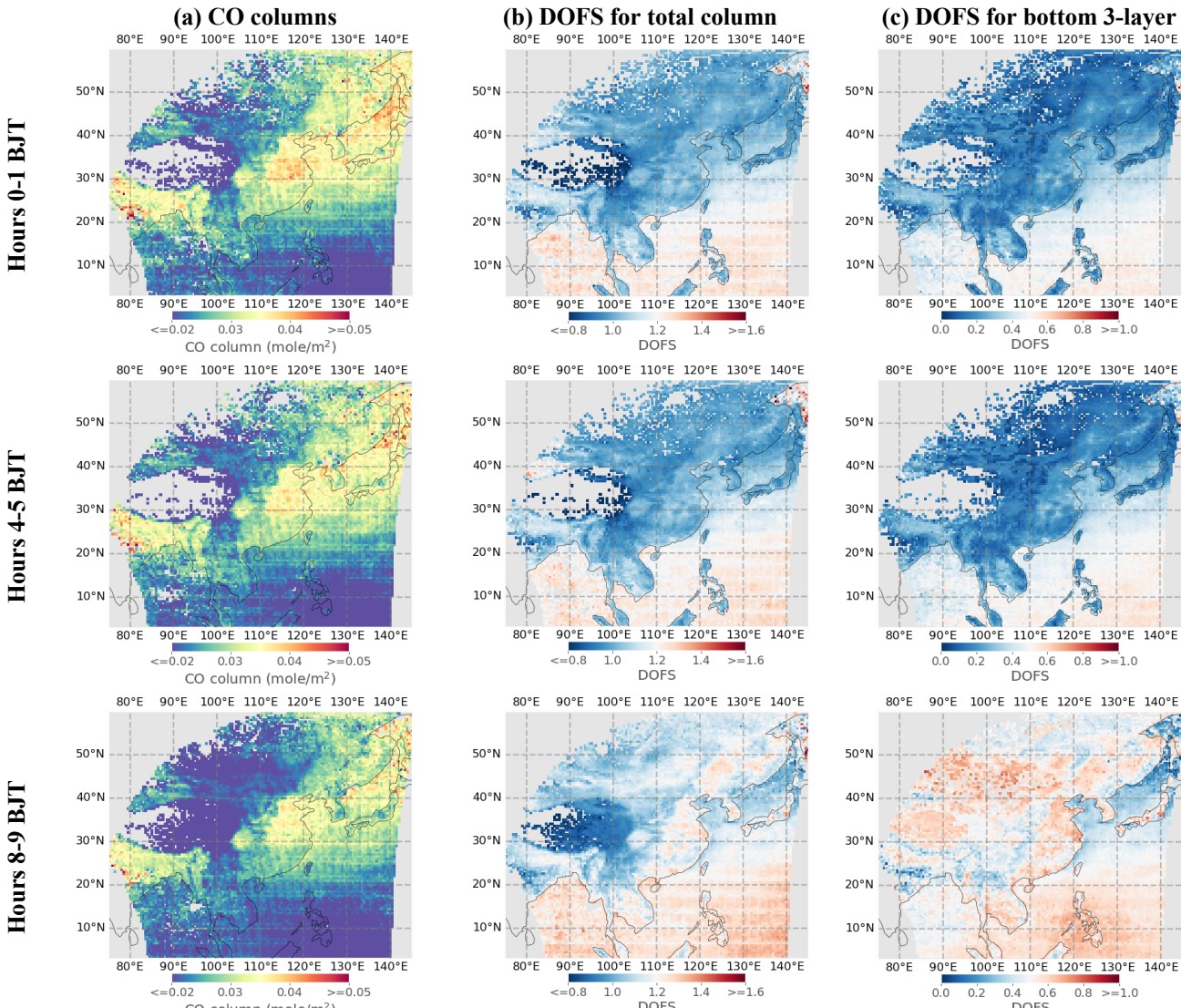

**Figure 11: (a) Maps of retrieved CO total columns from FY-4B/GIIRS for hours 0-1 (upper), hours 4-5 (middle), and hours 8-9 (lower) in Beijing Time (BJT; UTC+8). The CO total columns are monthly averages in July of 2022 that have been e-grided into 0.5°×0.5° grids; (b) The distribution of DOFS for the CO total column; and (c) The distribution of DOFS for the CO parital column of the bottom 3-layer (from the surface to 3km a.s.l.).**

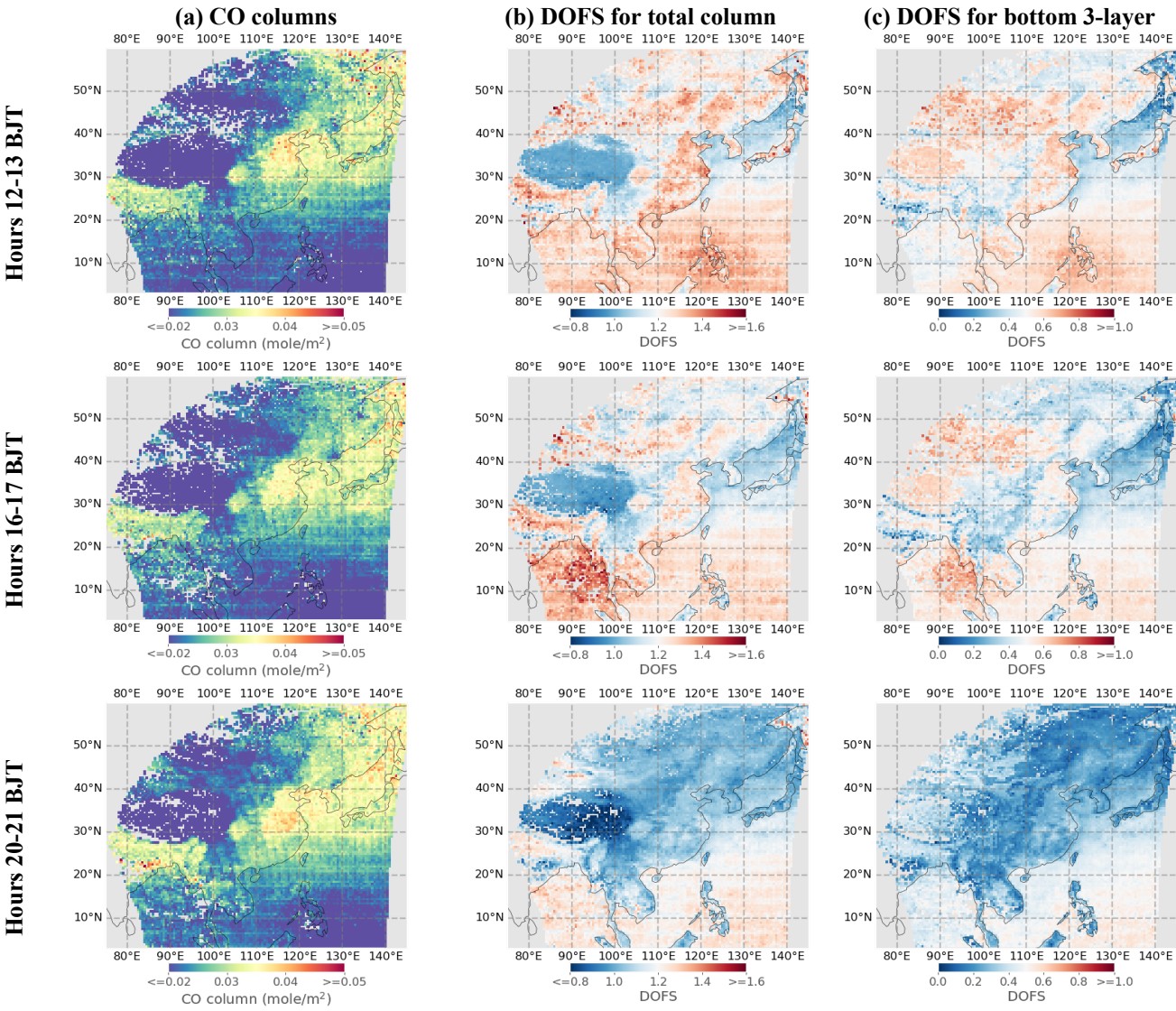

**Figure 12: The same as Fig. 10 but for hours 12-13 (upper), hours 16-17 (middle), and hours 20-21 (lower) in Beijing Time.**

## 6.4 Accuracy assessment by comparison with IASI CO retrievals

Different from FY-4B/GIIRS, IASI onboard Metop-B, which was launched in 2012, is a sun-synchronous polar-orbiting
infrared spectrometer designed to measure the upwelling spectral radiance in the infrared using a nadir viewing geometry, with
equator crossing times at 10:30 am and 10:30 pm LT, respectively, in the morning and evening. IASI has a dedicated CO
retrieval algorithm (**Hurtmans et al., 2012**) that was improved over time and has benefited from cross-comparisons with other
products (e.g., **George et al., 2009; Wachter et al., 2012; Worden et al., 2013; George et al., 2015**). Therefore, a comparison
would shed light on the difference between GIIRS infrared CO retrievals and the state-of-the-art retrievals from IASI. In this
section, we carry out the spatial and temporal comparisons using publicly available CO from IASI/Metop-B.

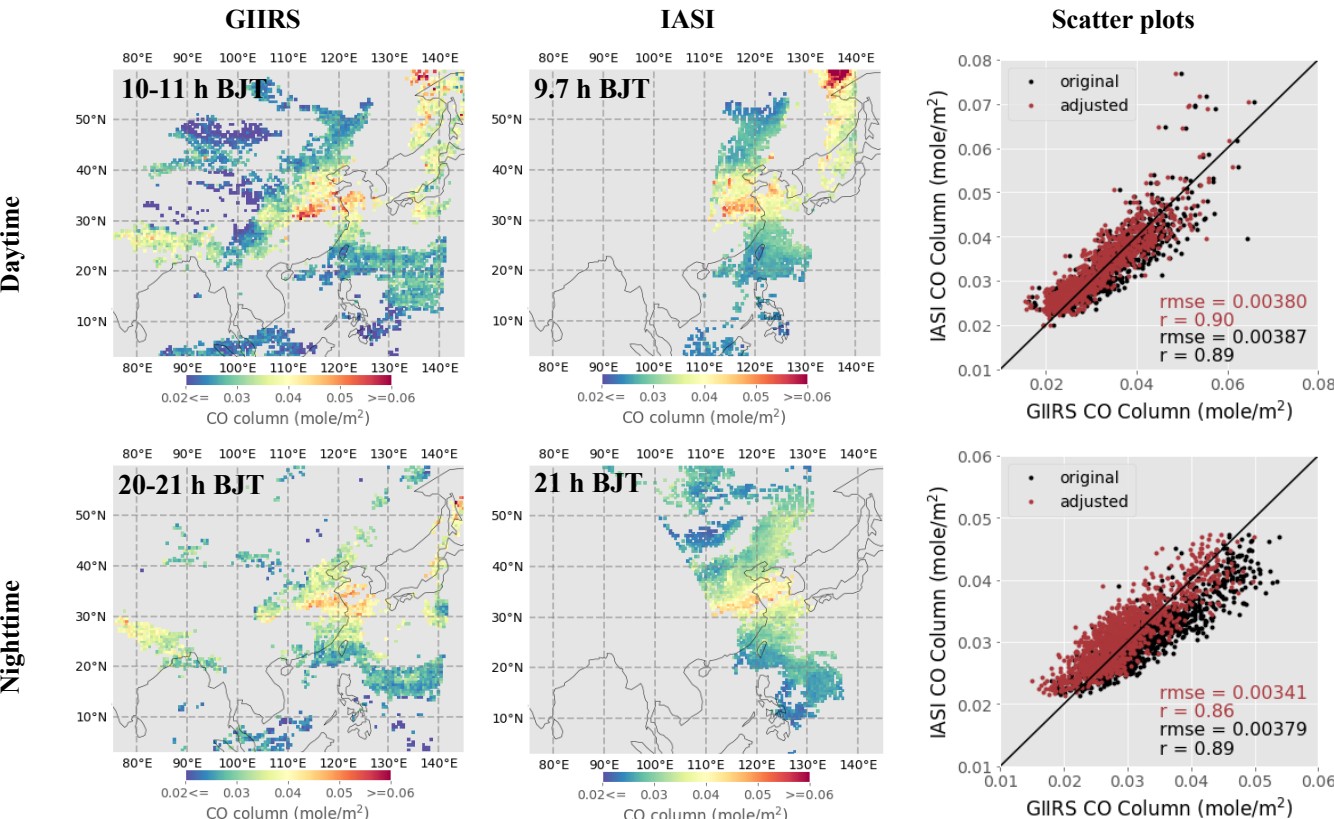

**Figure 13. Comparing CO column retrievals from GIIRS (1st column) and IASI (2nd column) on July 07, 2022, using daytime (upper panel) and nighttime (lower panel) retrievals. The observation hours are also indicated. The scatter plots (3rd column) show the comparison between GIIRS and IASI original and adjusted column data. The adjusted CO**
**column data are generated by adjusting the GIIRS CO retrievals based on the IASI a priori CO profile.**

For the spatial comparison, we use the daytime and nighttime observations on July 07, 2022 as examples. The observation hours of IASI (Supplementary **Fig. S7**) over the North China Plain are about 9.7h BJT for the daytime and 21h BJT for the

nighttime, which correspond to the GIIRS measurement cycles of 10-11h BJT and 20-21h BJT, respectively, for the daytime and nighttime. Because the GIIRS and IASI have different footprints and grid configurations, it is not straightforward to make point-by-point comparisons. We re-grid the retrieval data into 0.5°×0.5° grids and then make comparisons of the collocated grid data points. The results are shown in **Fig. 13**. We can see the spatial distribution of CO columns in the daytime is characterized by two source regions, the North China Plain as a source of anthropogenic emissions and the Siberia region as a

source of natural wildfire emissions. This wildfire over Siberia on July 07, 2022 is further confirmed using MODIS optical images and fire counts (not shown here). In the nighttime, the emissions from North China Plain persist. However, the wildfire regions are not covered. The scatter plots comparing the collocating observations show good agreements between the two datasets.

When comparing datasets from different instruments, data corrections including an a priori adjustment and an AK smoothing

are usually necessary to account for the difference in the a priori and AK (**Rodgers and Connor, 2003**). Fortunately, we found that the DOFS and the vertical sensitivity, as indicated by the AK diagonal elements, are similar between GIIRS and IASI, as shown in the **Appendix Figure A2 and Supplementary Fig. S8**. The DOFSs from both the daytime and nighttime retrievals are highly correlated although IASI DOFSs are higher by about 10%. The AK diagonal profiles from the examples shown in the **Appendix Figure A2(c)** and **(d)** are comparable between GIIRS and IASI for the lower atmosphere which accounts for

the majority of the total column. We therefore assume that both sensors have similar vertical sensitivity. Therefore, only the GIIRS data adjustment based on IASI's a priori CO profile is carried out. We adjust the GIIRS CO retrieval profile ($CO_{ret}^{GIIRS}$) to the IASI a priori profile ($CO_a^{IASI}$) by:

$$CO_{adj}^{GIIRS} = CO_{ret}^{GIIRS} + (A^{GIIRS} - I)(CO_a^{GIIRS} - CO_a^{IASI}) \, , \tag{6}$$

where $CO_{adj}^{GIIRS}$ is the adjusted profile result; $CO_a^{GIIRS}$ is the GIIRS a priori profile; $A^{GIIRS}$ is the GIIRS retrieval averaging

kernel matrix. **Fig. 13** (3ʳᵈ column) shows that this adjustment results in better agreements (smaller RMSE) between GIIRS and IASI. Furthermore, to test the impact of the small difference of AK (especially around 200 hPa) between GIIRS and IASI on the comparison, we applied the AK-smoothing to the IASI retrieved CO partial column profile retrievals (**Supplementary Text S1**). The results, as shown in **Supplementary Fig. S9**, from IASI and GIIRS are highly consistent, suggesting the small difference in AK between IASI and GIIRS are not significantly affecting the comparison of CO columns.

For comparing the time series of GIIRS and IASI, we focus on the four representative regions (North China Plain, Mongolia, East China Sea, and North India) and compare the regionally averaged CO total columns between GIIRS and IASI. The data processing is similar to the spatial comparison. First, the retrieval data are re-gridded into 0.5°×0.5° grids and the comparisons are made using the collocated grid data points to avoid bias related to uneven distribution of CO data points within the selected region. Then, according to the averaged observation hour of IASI in a specific region, the temporally closest measurement

cycle of GIIRS retrievals is used. Finally, the GIIRS CO retrievals are also adjusted to the IASI a priori following **Equation**

**(6)**. The comparison results of daily averaged CO columns are shown in **Fig. 14**. The statistics of the comparisons are shown in the **Supplementary Table S1**.

The direct comparison between GIIRS and IASI shows good agreement, with the majority of the correlation coefficients larger than 0.8. Specifically, the daily variabilities are highly consistent between the two datasets. Since in their retrieval algorithms the a priori CO total columns used in GIIRS and IASI do not vary with days, the daily changes of the retrieved CO total columns directly reflect the available information in the observations from both instruments. This consistency between GIIRS and the state-of-the-art IASI CO retrievals demonstrate the effectiveness of FY-GeoAIR algorithm in constraining CO profiles from GIIRS. Noted that in the nighttime, both instruments have weak sensitivity to the lower atmosphere, the agreement may just indicate that GIIRS and IASI have the same decreased sensitivity at nighttime. After correction using the a priori adjustment, we found that the changes are not uniform. While the consistencies for the nighttime cases improve for East China Sea and North India, it becomes worse for Mongolia, probably due to the larger retrieval uncertainty under lower temperature conditions in the nighttime that cause an increase in spectral noise.

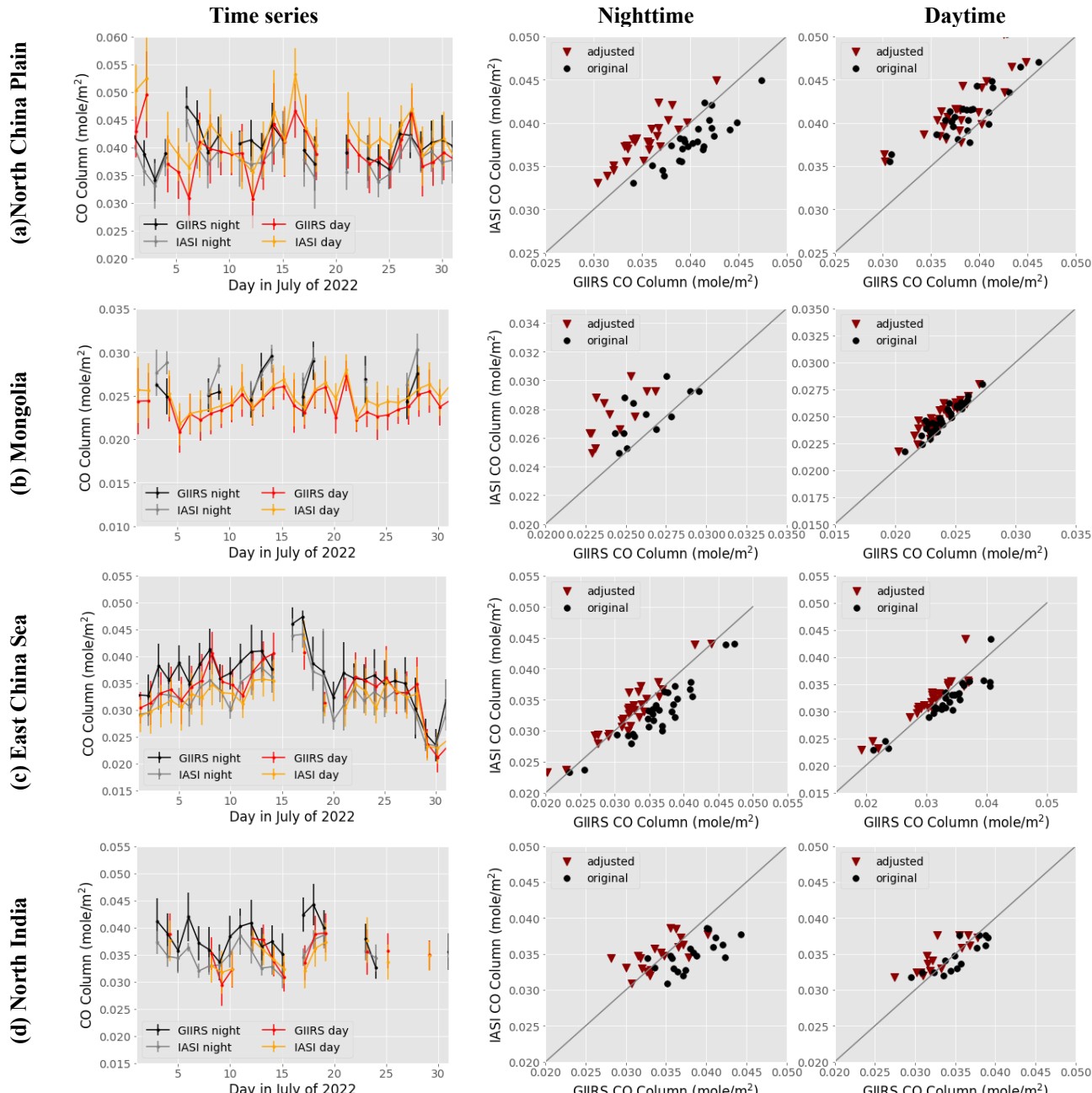

**Figure 14. Comparison of daily CO total column between GIIRS and IASI averaged over (a) North China Plain, (b) Mongolia, (c) the East China Sea, and (d) North India. The retrievals are re-grided to 0.5°×0.5° grids in the selected region. The daily mean value for each region is computed when at least 3 grid points are available. The first column shows their time series in July 2022. The second and the third columns are the scatter plots for the daytime and nighttime daily averages. The adjusted data are generated by adjusting the GIIRS CO profile retrievals based on the IASI a priori CO profile.**

## 7 Discussions

### 7.1 Applicability of the FY-4B/GIIRS spectra for retrieving CO columns in the winter season

For IR sounder like GIIRS, observations of low temperature targets are associated with high radiometric noise in brightness temperature, leading to unstable retrievals with large uncertainty. In the winter season when the surface and the atmospheric temperatures are dramatically reduced, the applicability of the GIIRS spectra in retrieving the CO columns is unknown. As a preliminary test, we apply the algorithm to observations in December when temperature is close to the lowest point. As shown in the supplementary **Fig. S10**, the average temperatures in North China Plain and East China Sea drop by about 20K, and in Mongolia the drop is more than 30K. Although the TC can be large over these regions, the high spectral noise related to low temperature may significantly reduce the detectivity. As a result, the RMSE of the fitting residual averaged over all retrievals in December is 1.16±0.82K, which doubles the estimate (0.63±0.13K) in July, as shown in the supplementary **Fig. S11**. The DOFS for the majority of retrievals in the high latitude regions has dropped to below 1.0, as shown in the supplementary **Fig. S12** with an example of CO column and DOFS maps in hour 3-4h BJT on December 18, 2022. If we apply the filter based on the RMSE of fitting residual <=1.0K, almost all the retrievals in the high latitude (>40°N) are filtered out. The availability of retrievals is much smaller for the night time observations due to its even lower temperature. Fortunately, since the sea surface temperature is relatively high in winter, the ocean regions in the low latitudes can still provide observations with low fitting residual and high DOFS that are comparable to summer retrievals.

### 7.2 Importance of a priori adjustment and AK smoothing for comparing retrievals

The importance of the AK matrix for intercomparison of retrievals has been described in detail by **Rodgers and Connor (2003)** using retrievals from MOPIIT, an LEO IR sounder, as an example of a space-born instrument. Because of the highly variable TC over the diverse land cover in East Asia, the GIIRS retrievals present distinctive vertical sensitivity for different hours at different locations, which makes the interpretation of the retrieval results more difficult and, therefore, better use of the AK matrix more important. For comparison with model simulations, as illustrated in the simulated synthetic experiments in **Sect. 5**, a correction as in **Eq. (5)** is necessary given the model simulated profile can be assumed to have uniform vertical sensitivity. For comparison with retrievals from other remote sounding instruments, which have different a priori and AK matrices, the a priori adjustment and AK-smoothing correction as in by **Rodgers and Connor (2003)** is necessary to reconcile the retrievals. Because of the mismatch in observation footprints and the heterogeneity of the AK matrix over land, the collocation of sounding measurements from different instruments should also be carefully implemented to make sure the comparison is not biased due to inappropriate spatial interpolation.

## 8. Conclusions

Using hyperspectral infrared measurements from GIIRS onboard FY-4B, we showed the first results of diurnal CO, an important trace gas in the atmosphere, measured from a GEO orbit. The performance of the algorithm is first evaluated by conducting retrieval experiments using simulated synthetic spectra. Retrieval results from one month of GIIRS spectra in July 2022 show that the DOFS for the majority is between 0.8 and 1.5 for the CO total column and between 0 and 0.8 for the bottom 3-layer (from the surface to 3km a.s.l.) atmosphere, which strongly depends on TC. Comparing the CO total column between GIIRS and IASI shows that the two datasets have good consistency in capturing the spatial and daily variabilities. This study demonstrates that the GIIRS retrievals are able to reproduce the temporal variability of CO total columns over East Asia in the daytime in July. Nevertheless, the retrievals have low detectivity in the nighttime due to their weak sensitivity to the ground level CO changes limited by low information content. Since CO plays an important role in tropospheric atmospheric chemistry and is an effective tracer of $CO_2$, the CO profile retrievals at a spatial resolution of 12 km and a temporal resolution of 2 hours from GIIRS have great potential in improving local and global air quality and climate research through model assimilation that takes into account the associated vertical sensitivity. The operational geostationary observation by GIIRS represents an important advancement over the once/twice-per-day observations provided by current LEO instruments.

In the coming future, CO observations from planning GEO missions, e.g., ESA's IRS and NASA's GeoCarb, along with GIIRS onboard future Fengyun satellite series, will greatly expand our capability in monitoring global CO emissions at high temporal resolution across Asia, Europe, and America. Moreover, combining NIR and TIR to measure CO could further improve constraining the CO profile from GEO orbits, as the NIR adds information in the boundary layer while the TIR is more capable of distinguishing near-surface and mid-troposphere (**Fu et al., 2016; Natraj et al., 2022**), which will be another very important advancement in the future.

**Data availability**

The CO retrieval data from FY-4B/GIIRS (https://doi.org/10.18170/DVN/M7DKKL) in this study are publicly available from the Peking University Open Research Data Platform at https://opendata.pku.edu.cn/; Future updates on FY-4B/GIIRS CO data will be posted on https://opendata.pku.edu.cn/dataverse/FYGEOAIR; FY-4B/GIIRS Level 1 data are publicly available from the FengYun Satellite Data Center at http://satellite.nsmc.org.cn/portalsite/default.aspx; IASI/Metop-B CO level 2 retrieval data are downloaded from IASI AERIs database portal at https://iasi.aeris-data.fr/co/; IASI is a joint mission of EUMETSAT and the Centre National d'Etudes Spatiales (CNES, France). The authors acknowledge the AERIS data infrastructure for providing access to the IASI data in this study, ULB-LATMOS for the development of the retrieval algorithms, and Eumetsat/AC SAF for CO/O3 data production; The surface emissivity datasets are downloaded from the Global Infrared Land Surface Emissivity: UW-Madison Baseline Fit Emissivity Database at https://cimss.ssec.wisc.edu/iremis/; The ECMWF ERA5 reanalysis datasets are available from the Copernicus Climate Data Store at https://cds.climate.copernicus.eu/; The ECMWF atmospheric composition datasets are available from the Copernicus Atmosphere Data Store at https://ads.atmosphere.copernicus.eu/.

**Acknowledgment**

Z.-C. Zeng acknowledges funding from the National Natural Science Foundation of China (grant no. 42275142 and no. 12292981), the National Key R&D Program of China (grant no. 2022YFA1003801), and the Fundamental Research Funds for the Central Universities at Peking University (grant no. 7101302981).

**Author contribution**

Z.Z. designed the study, developed the forward model and retrieval codes, carried out the experiments and results analysis, and prepared the manuscript. L.L. and C.Q. provided guidance on using the FY-4B/GIIRS L1 spectra data and carried out experiments for assessing spectra uncertainty. All authors reviewed the manuscript.

**Competing interest**

The authors declare that they have no conflict of interest.

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

**Appendix**

**(a) CO bottom-up inventories**

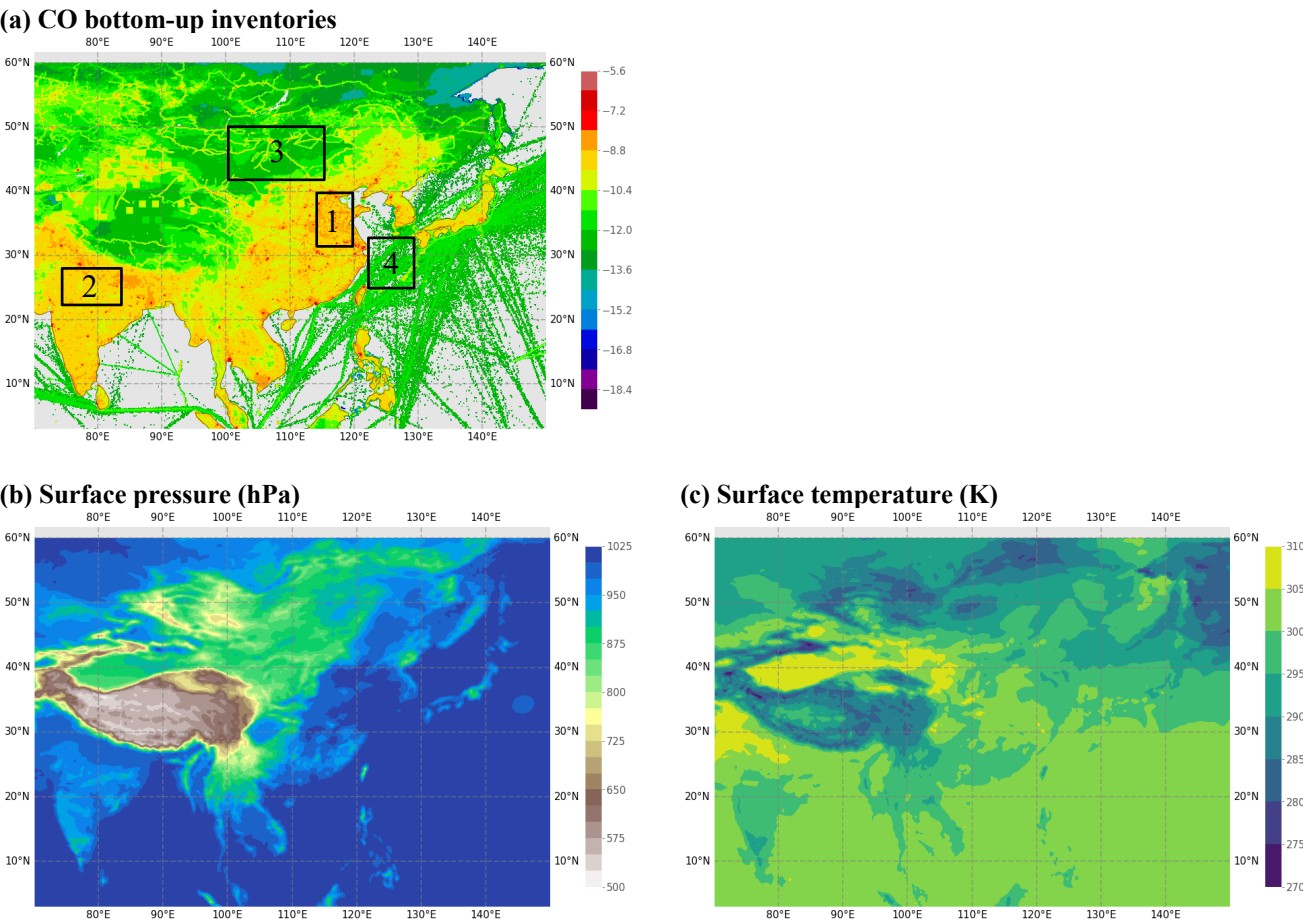

**Figure A1: (a) Bottom-up estimated CO emissions on July 07, 2022, from the CAMS model. These emissions include anthropogenic CO emissions from fossil fuel use on land, shipping, and aviation, and natural CO emissions from vegetation, soil, the ocean, and termites. The emissions are in a unit of [kg m$^{-2}$ s$^{-1}$] at log$_e$10 scale; Four representative regions in black box are selected for inter-comparisons, including (1) North China Plain (covering 32°-40°N and 114°-120°E), which represents industrialized urban regions with persistently high CO emissions; (2) Mongolia (covering 42°-**

**50°N and 100°-115°E), which represents CO background regions; (3) the East China Sea (covering 25°-33°N and 122°-129°E), which represents ocean surface; (4) North India (23-28°N and 75°-83°E), which represents urban CO source region in India. (b) Surface pressure from ECMWF ERA5 reanalysis for the surface layer on July 07, 2022; (c) The surface temperature from ECMWF ERA5 reanalysis for the surface layer on July 07, 2022.**

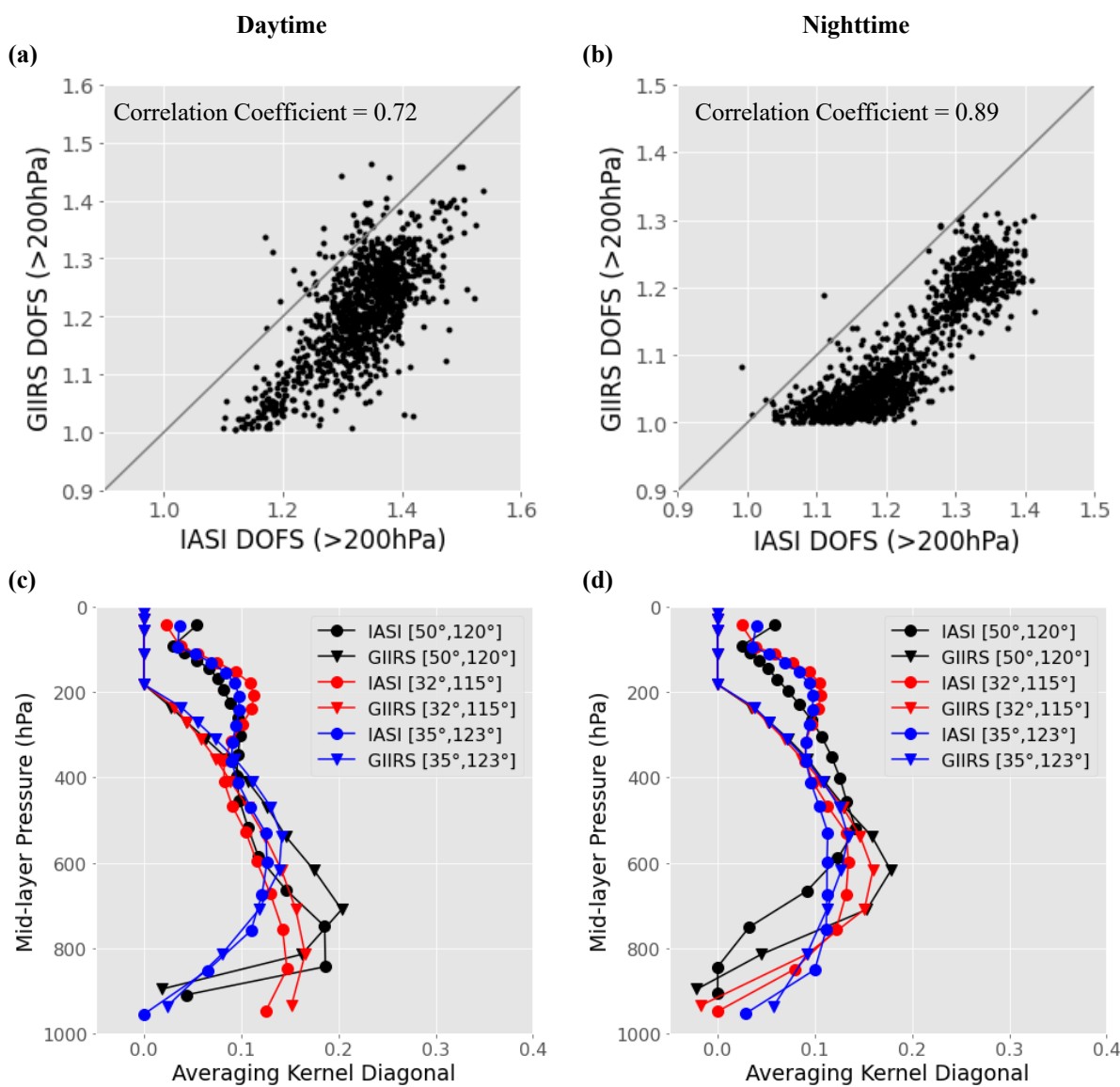

**Figure A2. Comparison of GIIRS and IASI DOFS for retrievals shown in Fig. 13. The DOFSs are re-grided into 0.5°×0.5°**
**grids and the collocated grid data are compared for the daytime observation in (a) and nighttime in (b). A comparison**
**of detectivity indicated by averaging kernel diagonals are shown in (c) and (d) from three selected locations, including**
**observations in Mongolia in black, in North China Plain in red, and in East China Sea in blue.**