# Peer review of "Diurnal carbon monoxide observed from a geostationary infrared hyperspectral sounder: First result from GIIRS onboard FengYun-4B"

_Atmospheric Measurement Techniques, 2022_

## Author Comment (AC1)

**Response to Reviewer #1**

We thank the reviewer for his/her constructive comments and suggestions to improve the quality and clarity of our manuscript. We have made major and careful modifications to the original manuscript according to all the comments and suggestions from the reviewers. The major modifications include:

(1) We have changed to using a fixed CO profile as the a priori for all the retrievals, and make changes to all relevant figures. The use of a fixed a priori make it easy to interpret and compare with models and detect anomalies such as wildfires emissions. Specifically, for our purpose of retrieving the diurnal changes of CO columns, the main topic of this study, a fixed a priori is preferred because any significant perturbation to the constant a priori, which does not change diurnally, may indicate information that is retrieved from the observed spectra;

(2) We removed the 0-1km results and replaced it with the bottom 3-layer, which ranges from the surface to 3km above sea level. In addition, we added North India as the 4[th] representative region in the analysis, besides North China Plain, Mongolia, and East China Sea.

(3) We added diurnal cycle comparison of the retrieved CO columns and DOFS, boundary layer height from ERA5 reanalysis, and model simulations of CO columns from CAMS EAC4 reanalysis. All time in the revised manuscript has been changed to Beijing Time (UTC +8).

(4) A detailed comparison between our retrievals and IASI has been added, including spatial and temporal comparisons. We showed that the DOFS between GIIRS and IASI are comparable, and the vertical sensitivities, as quantified by the AK matrix, are also similar between GIIRS and IASI.

(5) We added in the Discussions section about the applicability of the algorithm in retrieving CO using GIIRS observation in the winter season, and used December observations as examples.

Item-by-item responses to the specific comments are provided below, in which the reviews' comments are in blue, our responses in **black**, and modifications of the original manuscript are indicated by highlight in yellow in the revised manuscript.

**Review of Zeng et al., 2022**

This paper describes CO retrievals from the geostationary hyperspectral infrared sounders GIIRS onboard FY-4B. This is the first publication presenting CO retrievals from a geostationary platform that could be valuable to document the diurnal cycle of this species in the lower troposphere. The paper is correctly writen and structured with some interesting information. It is therefore adapted to AMT.

Nevertheless I have some important concerns and questions that have to be adressed before publication. My most important concern is about the diurnal cycle itself which is

the main topic of the paper and included in its title. It should be better documented and compared to other datasets to be validated at minima.

Thank you for your positive comment and constructive suggestions. We have made major changes related to the analysis of the diurnal cycle of CO column retrieval, including (1) We have added Fig. 8 and Fig. 10 to compare the diurnal cycles of the retrieved CO columns, DOFSs from the retrievals, PBLH from ERA5 reanalysis, and model simulations from ECMWF EAC4; (2) In Fig. 13 and Fig. 14, we compare our CO column retrievals with IASI retrievals, especially the day-night contrast of CO columns; (3) In our synthetic experiment as shown in Fig. 4, we separated the results into hours to investigate the accuracy of the retrieval algorithm in reproducing the diurnal change of CO columns; (4) In Section 6.3, the DOFS for the total column and the bottom 3-layer are shown separately to examine how the retrievals are sensitive to the total column and the lower atmosphere. For details, please see our responses to your specific comments below.

**CO a priori profiles :**

The advantages and inconvenients of a climatological a priori are mentioned in section 7.1 and the use of a single a priori as a way to improve the algorithm to detect anomalies. It should also be mentioned that using such an a priori makes the retrievals more complicated to interpret and to use for model validation.

The use of a 3 hourly profile climatology based on 5 years simulation is done to help to provide the correct diurnal cycle to the retrieval algorithm. But CO is a pollutant with a lifetime much larger than a day. The daily cycle for CO is not as important as for NOx. The authors should provide the plots of the daily variations of CO in the troposphere and lower troposphere in the 3 selected regions from the ECMWF CAMS for instance together with the surface and bottom air temperatures in Fig 2.

Figure 4 shows that the same low biases for high concentrations in the a priori are partially kept in the retrieval for North China Plain and Mongolia. The dissapearence of these biases when the AvKs are applied to the « true » profiles clearly indicates that these biases are linked to the a priori and the lack of sensitivity of the sensor/retrieval to the polluted BL. As stated by the authors, this problem could be related to the too tight a priori covariance matrix used with the climatological profiles but it is not sure. It would be very interesting to provide some results from a simple test using a single a priori profile and its more loose a priori covariance to verify this assumption. In that case the signal to noise ratio for the retrieval which has been tuned according to the a priori covariance matrices (section 6.1) should be lowered which could lead to a destabilisation of the retrieval and possible oscillations in the profiles.

We thank the reviewer for the very constructive suggestions. We have made substantial revisions according to your suggestions:

(1) In the revised manuscript, we changed the a priori from time-varying profiles to a fixed CO profile derived from model simulations. The reason, as you mentioned, is that a time-varying a priori makes the retrieval results more complicated to interpret and to use for model validation. Also, the small errors with the time-varying a priori profiles also make it difficult to detect anomalies from unexpected events such as wildfire emissions. For our purpose of retrieving the diurnal changes of CO columns, the main topic of this study, a fixed a priori is preferred because any significant perturbation to the constant a priori, which does not change diurnally, may indicate information that is retrieved from the observed spectra. The updated a priori CO profile and its covariance matrix are shown in Fig. (2). As expected, the DOFSs are higher for the retrievals using fixed a priori (with a larger variability) compared to the time-varying profiles. Compared to IASI, we found our results show a very good agreement for the urban source regions as well as natural wildfire emissions, as shown in Fig. 13 and Fig. 14 in the revised manuscript;

(2) The averaged daily cycle of CO columns and the ground-level CO concentration are shown in Figure 10(c) and the supplementary Fig. S6. They are used to compare with the diurnal change of the retrieved CO columns, and DOFS from the retrieval and the boundary layer height as shown in Fig. 8 and Fig. 10, respectively.

(3) Using the fixed a priori CO profile, we have repeated the simulation experiment and updated Fig. 4. We found that in the daytime when DOFS is large the retrieval results have a good agreement with the assumed "truth". This suggests that a single a priori profile with a loose a priori covariance used in this study (Fig. 2) improves the retrieval results compared to previous results using the time-varying profiles but with a tight a priori covariance.

Related statements have been added to the revised manuscript in Section 3.2, Section 5, and Section 6.3.

**What diurnal cycle ?**

The problem to document the CO diurnal cycle with the GIIRS retrievals come from the fact that it could be linked to :

- the real CO cycle that is the objective

- the variability of the BL layer with probably a better detection of pollution in the afternoon when the BL is higher where the sensor is more sensitive

- the variability of the DOFS

In order to disantangle these different sources of diurnal CO cycles

- the diurnal cycles of CO total columns over the 3 selected zones should be provided clearly the same way as the DOFS in Fig 8. The plot of the DOFS for the 0-1 km could be

removed as the retrieval for this layer provides no relevant information (see next comment) and as its diurnal variability is mostly similar to the total column.

- the diurnal cycle of the BL height could also be documented from ECMWF ERA5 data for instance.

- the diurnal cycle of CO from other sources such as local pollution networks in China/Beijing area, ECMWF CAMS used for the a priori (see above) and some references to relevant publications should be provided to check wether or not GIIRS retrievals are sensitive to BL pollution diurnal variability.

Thank you for your thoughtful and very constructive suggestions. We have made the following revisions according to your suggestions:

(1) In Section 6.3, we added what you have just mentioned to explain what may drive the diurnal changes of the CO column retrievals: "**The diurnal changes of CO from FY-4B/GIIRS retrievals may be linked to (a) the real CO variabilities, the objective of this study, driven by emissions and transport, and (b) the change in the detectivity by the instrument as reflected in the DOFS from the retrieval algorithm, which can be affected by the change in TC and CO concentration.**"

(2) As an attempt to disentangle these different factors, in Fig. 8, we show the diurnal changes of the DOFS for both the total column and the bottom 3-layer (from the surface to 3km a.s.l.), the CO columns from retrievals, the boundary layer height (BLH) from ECMWF reanalysis data, and the ECMWF CAMS EAC4 CO simulations over the 4 selected regions. The 0-1km figures have been removed in the revised manuscript. The comparison results suggest that a direct interpretation of the authentic diurnal column variabilities from the retrieved CO columns is challenging given the entangled effects of the real CO changes and the variable detectivity. A better solution is to use mode assimilation (e.g., EAC4) that takes into account the retrieved CO profile and the vertical sensitivity in order to disentangle the different contributions. Please see the details in Section 6.3 in the revised manuscript.

(3) Unfortunately, the most recent data from local measurement network are not available for the public yet. Instead, we use model simulations from CAMS EAC4 that has assimilated satellite observations (IASI and MOPPIT) for the comparison, as shown by the CO column in Fig. 9(c) and the ground-level CO concentration in Fig. S6. From the model simulations, which has assimilated satellite observations of IASI and MOPPIT, we see the change in total columns in all selected regions are very small (less than 2% on average) which can be primarily attributed to the diurnal change in BLH. In the supplementary Fig. S6, the model simulated ground-level CO concentrations show a much larger variation compared to the CO columns. However, model simulations from EAC4 have large grids (0.75°×0.75°) and low temporal resolution (3-hour) which are not sufficient to resolve the local CO changes that are usually measured over the urban centrals. To the first order, the change in the ground-level CO concentration averaged

over the selected region is primarily driven by the BLH change, and the traffic emission peaks are not discernable from the time series.

(4) Moreover, we collected information about the CO diurnal cycle in China and India from published papers. Please see our response to your next comment.

I have some doubts about the diurnal variability displayed in fig 10 and 11 in the NCP : the maxima are detected between 16 and 22 UTC that is between midnight and 6AM Beijing time (If I understood correctly). So it does not correspond to the time of day (i) with the highest activity where we expect the largest emissions and concentrations (this should be highlighted by surface /CAMS data as proposed above) (ii) with the highest BL which is in the afternoon (iii) with the largerst DOFS which is the begining of the afternoon (see Fig 8 and Fig 10 b). The authors have to provide some explanations about this peculiar diurnal cycle.

A typical diurnal change of ground-level CO, like what you described, is directly affected by the diurnal emission pattern from traffic besides BLH change. In urban regions, ground level CO concentrations from in-situ ground-based observations show a distinctive double-peak diurnal cycle corresponding to the morning and evening traffic rush hours. In the morning, the increase in traffic emissions results in the morning peak; As BLH gets larger, air dilution takes place and the concentration drops; In the evening, traffic emissions increases while the BLH quickly decreases which results in an evening peak. These diurnal pattens have been observed cross many cities in Asia (e.g., Ran et al., 2009; Chen et al., 2020; Meng et al., 2009; Verma et al., 2017).

However, the diurnal changes of CO columns and the surface layer concentration are not expected to be the same. As shown in Stremme et al. (2009), which retrieved diurnal CO column changes in the Mexico City using ground-based solar and lunar infrared spectroscopy, found that the diurnal changes in the total CO column and the surface level concentration can be very different. The total CO column within the city presents large variations with contributions from urban CO emissions at the surface and the transport of cleaner or more polluted air masses into the study area.

The diurnal cycle of the retrieved CO columns from FY-4B/GIIRS, shown in Fig. 10(a), presents impacts by the diurnal change of DOFS. Noted that the a priori CO columns averaged for the year of 2021 are 0.038, 0.028, 0.039, 0.036 mole/m2, respectively, for North China Plain, Mongolia, East China Sea, and North India. Since the DOFS for the nighttime is low, especially for the lower atmosphere, the nighttime column retrievals generally tend to close to the a priori columns, resulting in a bow shape. In the daytime, the CO column retrievals are well constrained by the observations. Since the summer time CO is generally lower than the yearly mean (Chen et al., 2020) based on CO's seasonal cycle, we see the column retrievals averaged in July are generally lower than the a priori column value which is derived from annual mean of model simulations. These results suggest that a direct interpretation of the authentic diurnal column variabilities from the retrieved CO columns is challenging given the entangled effects of the real CO changes and the variable detectivity. A better solution is to use mode assimilation (e.g.,

EAC4) that takes into account the retrieved CO profile and the vertical sensitivity in order to disentangle the different contributions.

We have added the above statements in the revised manuscript, please see Section 6.3.

**Why 0-1 km layer ?**

The 0-1 km could be interesting to document BL pollution but it is characterised by a very low DFS mostly below 0.1 to 0.15 (Fig 3 and 9) and below 0.125 on Fig 8 which means that there is almost no information about this layer in the retrieval whatever the thermal contrast. DOFS is even negative (what does that means?) in Fig 8 and 9 showing that this layer is absolutely not a good choice.

The sentence line 400 « The DOFS can be as large as 0.3 providing a strong constraint on the bottom 0-1 km » is a flagrant overstatement (just a couple of points at 0.3 in Fig 9!!!) and should be removed or changed. Even a DOFS of 0.3 would have meant that the information for this layer is low.

In Figure 7 that displays the AvKs we see that the AvKs peak at 800 or 700 hPa in the best cases.

I therefore do not see the relevance to display results about the 0-1 km layer in the different figures. As the DOFS for the total columns are roughly between 0.8 and 1.2, the authors should separate the atmosphere/troposphere in the two layers in which the information is equally provided and display results for the lowermost of those layers.

Thanks for your thoughtful suggestions. We have removed the 0-1km results and replaced with results of the bottom 3-layer, which ranges from the ground to 3km above the sea level. As shown in Fig. 8, in the daytime, the bottom 3-layer has a DOFS that is about half of the total column. This separation of the upper and lower atmosphere also corresponds to the peak sensitivity in Fig. 7 as you mentioned. Related changes have been made in Fig. 3, Fig. 8, Fig. 9, Fig. 10 and Fig. 11. We have also removed the statement that overstate the DOFS.

**Comparisons with IASI :**

The comparison with IASI data is made to partly validate the diurnal variations but some important information is missing :

- as there are only two overpasses of IASI daily at 9:30 LST AM and PM, the authors have to detail how they average the GIIRS data temporaly which is not clear at all.

- the correlation coefficients and rmse are given in the Fig 12 but a table with those figures and other basic statistics such as mean biases +/- rmsd should be added.

- the comparison methology to smooth IASI with GIIRS AvKs is assuming that IASI has a much better vertical resolution than GIIRS which is not the case (IASI has probably a DOFS of 1.5). In that sense it is worth to display IASI AvKs to compare with GIIRS (as in Fig 7) and to provide IASI's DOFS. It would probably be better to avoid to apply equation 11 assuming that both sensors have similar vertical sensitivity or to use the more (too) complicated methodology detailed in Rodgers and Connor, JGR (2003).

- IASI CO "diurnal cycle" is mostly related to its decreased sensitivity at night. So the agreement with GIIRS for day and night described as good by the authors is just indicating that GIIRS has the same decreased sensitivity at night. There should be some statements about this issue.

Thank you for your great suggestions. We have made changes related to the comparison with IASI:

(1) Besides the comparison of daily time series in July, we have added Fig. 13 and selected July 07 as an example to compare the spatial distribution between GIIRS and IASI, which shows a good agreement between the two datasets.

(2) We have added more details about how the comparison was made. For the spatial comparison, we added **"For the spatial comparison, we use the daytime and nighttime observations on July 07, 2022 as examples. The observation hours of IASI (Supplementary Fig. S7) over the North China Plain are about 9.7h BJT for the daytime and 21h BJT for the nighttime, which correspond to the GIIRS measurement cycles of 10-11h BJT and 20-21h BJT, respectively, for the daytime and nighttime. Because the GIIRS and IASI have different footprints and grid configurations, it is not straightforward to make point-by-point comparisons. We re-grid the retrieval data into 0.5°×0.5° grids and then make comparisons of the collocated grid data points."** For the time series comparison, we added "**For comparing the time series of GIIRS and IASI, we focus on the four representative regions (North China Plain, Mongolia, East China Sea, and North India) and compare the regionally averaged CO total columns between GIIRS and IASI. The data processing is similar to the spatial comparison. First, the retrieval data are re-grided into 0.5°×0.5° grids and the comparisons are made using the collocated grid data points to avoid bias related to uneven distribution of CO data points within the selected region. Then, according to the averaged observation hour of IASI in a specific region, the temporally closest measurement cycle of GIIRS retrievals is used. Finally, the GIIRS CO retrievals are also adjusted to the IASI a priori following Equation (6). The comparison results of daily averaged CO columns are shown in Fig. 14. The statistics of the comparisons are shown in the Supplementary Table S1.**"

(3) Supplementary Table 1 is added to provide statistics of the comparison, including correlation coefficient, RMSE, mean bias and the error standard deviation.

(4) Since we have used a fixed CO profile as the a priori, we have re-evaluated the comparison using a priori adjustment and AK smoothing. Our results show that GIIRS retrievals with a fixed CO profile as the a priori have sensitivity (reflected by the AK diagonal) that is similar to IASI (Appendix Fig. A2). We therefore only apply the a priori adjustment to the GIIRS data using the IASI a priori, but not the AK smoothing assuming that the two sensors have similar vertical sensitivity as you mentioned.

(5) We have added the statement to Section 6.4: "**Noted that in the nighttime, both instruments have weak sensitivity to the bottom atmosphere, the agreement may just indicate that GIIRS and IASI have the same decreased sensitivity at nighttime.**"

For details, please see Section 6.4 in the revised manuscript.

Detailed comments :

Section 3 and 4 :

Some generalities about radiative transfer and retrieval methodology and basic known equations could be removed. Equations 3 to 8 have been largely documented such as in Rodgers (2000) and it is unnecessary to repeat this here.

We have removed Equations 3, 5, 6 and 7, but keep Equations 4 and 8 as they are important to the description of the observation characteristics and the settings of the retrieval algorithms.

Time : the time is given in UTC but it is not a correct choice to interpret diurnal cycles around China. Beijing local solar time would be much better. Furthermore the time is often given without the precision that it is UTC.

We have changed the time used in the paper to Beijing Time, which is UTC+8.0.

Figure 7 : precise the time system chosen.

We have changed it to Beijing Time (UTC+8)

Figure 8 : we suppose it is UTC !

We have changed it to Beijing Time (UTC+8)

Figure 10 : please provide hour in LST because UTC is not adapted to the geographical zone.

We have changed it to Beijing Time (UTC+8)

Fig 12 : Dayth => Day

 Changed.

Line 59 : The authors mention Kobayashi et al. (1999) as one of the first attempt to document CO from space with the japanese IMG ADEOS. Nevertheless, this paper do not present CO retrievals from IMG. The only retrievals of CO from this first spaceborne IR FTS have been published later by Barret et al. (2005).

Thanks. We have replaced "Kobayashi et al. (1999)" by "Barret et al. (2005)", and Kobayashi et al. (1999) is move to be referenced for IMG.

**refs:**

C.D. Rodgers and B.J. Connor, Intercomparison of remote sounding instruments, JGR, 2003.

B. Barret, et al., Global carbon monoxide vertical distributions from spaceborne high-resolution FTIR nadir measurements, ACP, 2005.

We have added the second reference. The first one was already cited in the original manuscript.

---

## Author Comment (AC2)

**Reviewer #2**

We thank the reviewer for his/her constructive comments and suggestions to improve the quality and clarity of our manuscript. We have made major and careful modifications to the original manuscript according to all the comments and suggestions from the reviewers. The major modifications include:

(1) We have changed to using a fixed CO profile as the a priori for all the retrievals, and make changes to all relevant figures. The use of a fixed a priori make it easy to interpret and compare with models and detect anomalies such as wildfires emissions. Specifically, for our purpose of retrieving the diurnal changes of CO columns, the main topic of this study, a fixed a priori is preferred because any significant perturbation to the constant a priori, which does not change diurnally, may indicate information that is retrieved from the observed spectra;

(2) We removed the 0-1km results and replaced it with the bottom 3-layer, which ranges from the surface to 3km above sea level. In addition, we added North India as the 4th representative region in the analysis, besides North China Plain, Mongolia, and East China Sea.

(3) We added diurnal cycle comparison of the retrieved CO columns and DOFS, boundary layer height from ERA5 reanalysis, and model simulations of CO columns from CAMS EAC4 reanalysis. All time in the revised manuscript has been changed to Beijing Time (UTC +8).

(4) A detailed comparison between our retrievals and IASI has been added, including spatial and temporal comparisons. We showed that the DOFS between GIIRS and IASI are comparable, and the vertical sensitivities, as quantified by the AK matrix, are also similar between GIIRS and IASI.

(5) We added in the Discussions section about the applicability of the algorithm in retrieving CO using GIIRS observation in the winter season, and used December observations as examples.

Item-by-item responses to the specific comments are provided below, in which the reviews' comments are in blue, our responses in **black**, and modifications of the original manuscript are indicated by highlight in yellow in the revised manuscript.

This study describes the CO retrieval algorithm for the geostationary GIIRS sounders on board FY-4B satellite. The paper is in general well explained and clear, particularly for the algorithm, however some information or clarification are missing.

Comments:

Ln 9: The year of when the sounder was launched should be indicated in the abstract to inform the reader.

We have added the information in the abstract: **"… using observations from GIIRS onboard FY-4B, which was launched in June 2021, …"**

Ln 10: how having hyperspectral measurements of CO provide diurnal observation of CO? I would think that having geostationary CO data allow observation of diurnal CO variability. Could you reformulate your sentence or give more precision?

We have rephrased this statement: "**With hyperspectral measurement collected from a geostationary orbit …**"

Ln 34. I would maybe say "(GEO) orbit can provide contiguous coverage with similar or higher spatial resolution than LEO and a revisit time of 1-2 hours [...]" or a similar sentence. Because, GIIRS has the same spatial resolution than IASI (12km diameter at nadir).

We have changed the statement as you suggested: "… **with similar or higher spatial resolution …**"

Ln. 89: Figure A1. b and c, are these figures for the same day as Fig. A1.a?

Yes. We have added the date information in the figure caption: "… **on July 07, 2022**".

Fig. 1b. The values on Fig. 1B are a too small.

We have enlarged the font size of Fig. 1B. The number index (from 1 to 128) of the detector array pixel is unchanged since these index number is not key information.

Fig. 1c. It would be interesting to have, as well, the Jacobian by pressure (the ones used for the radiative transfer model). This would inform on the variability that GIIRS channels sensitivity have depending on atmospheric pressure. Additionally, there is no comment on this figure. What do you conclude with this figure in term of sensitivity for CO and $H_2O$ with GIIRS?

We have added Figure S1 in the supplementary materials that shows an example of Jacobian as a function of pressure for three channels: strong CO absorption channel at 2165.625 cm$^{-1}$, median absorption channel at 2166.250 cm$^{-1}$, and weak absorption channel at 2166.875 cm$^{-1}$.

The added the following statement as conclusions from Fig. 1(c) and Fig. S1: "**In this micro-window, the absorption features from CO and the primary interference gas $H_2O$ are mostly separated and can be distinguished. Examples of Jacobian as a function of pressure and absorption strength are shown in the supplementary Figure S1. The changing Jacobian values demonstrate the vertical sensitivity of the CO absorption lines at different pressure levels, which peaks at the surface layer in the daytime and at mid-troposphere in the nighttime.**"

Ln. 171. You computed 3-hourly CO profile climatology for each month, which month are you talking about? The period of study has not been introduced yet, except result for July 2022 introduced in the abstract.

In the original manuscript, the climatology was constructed for July only, and we have only retrieved data in July. In the revised manuscript, we have changed to using a fixed CO profile as the a priori, instead of a time-varying a priori CO profile. The fixed CO a priori profile is derived from CO simulations from CAMS EAC4 reanalysis. All 3-hourly simulation results in 2021 are used. The details are described in Section 3.2.

As a response to your question in the end about whether we have applied the retrieval algorithm to another month, we have tested the retrieval algorithm using observations in December 2022, when the temperatures of the surface and the atmosphere reach to the lowest point in a year. Figures S9, S10, S11 are the results added. Please see our detailed response to your comment below.

It has been changed: **"In the spectral window from 2143 cm$^{-1}$ to 2181.25 cm$^{-1}$"**

Ln. 194. "The number of pressure grids in the forward RT model should be large enough to reduce the error [...]". A reference is missing here regarding this remark.

The reference Clough et al. (2016) has been added for this statement.

Ln 262: To be consistent, I would write the title as " Averaging kernel (AK) matrix and Degree of Freedom for Signal (DOFS)"

It has been changed as suggested.

Ln. 276. How much data are removed after the quality filter? Similarly, how much data are removed before and after labeled clear sky for the period of your study, before the post-screening is done?

For the cloud screening, in Section 2.3, we added:

**"For each measurement cycle, there are 12×27 FORs and each FOR collects 16×8 observations using the infrared plane array detector. In total, there are 41472 observations. For each day with 12 measurement cycle, the total observation is about 500K. After cloud screening and excluding data with viewing zenith angle larger than 70°, the average daily observation number for clear sky is about 90K in July of 2022."**

For the post-screening, in Section 4.3, we added:

**"In the post-processing, multiple filters are applied to ensure good retrieval quality. First, retrievals that fail to converge after 10 iterations are excluded. Second, retrievals with the**

**goodness of fit, quantified by reduced $\chi^2$, less than 1.5 are excluded. Lastly, Retrievals with root-mean-square-error of fitting BT residual that are more than one standard deviation away from the mean, which is about 0.7K for July 2022, are excluded. After data screening, the total number of observations, which is 2,812,071, is reduced to 2,045,228 in July 2022."**

Ln. 289. Could you precise why you added a Gaussian white noise? Is the added noise mentioned Ln 291 referring to the Gaussian white noise. If yes, then Ln 291 should appear just after the white noise is mentioned Ln. 289.

The Gaussian white noise is noise with mean of zero and, in this study, a standard deviation of NedR×1.5. This type of noise is usually assumed for the measurement spectra assuming they are not biased (mean=0) but has a random noise level that following Gaussian distribution. We have rephrased the statement to be **"… Gaussian white noise with mean of zero and a standard deviation equal to NedR×1.5."**

Also, the sentence referring to the noise has been moved forward to be right after the description of the white noise.

Ln. 293. You could introduce a map to visualize the regions of interest.

We have labelled the four regions of interest in Figure A1(a) with explanations in the figure caption.

Ln. 304. How can you conclude that from Figure 2?

In the revised manuscript, we have added the diurnal change of DOFS. This sentence has been rephrased to be: **"The complexity of the diurnal TC change as demonstrated by various land use types in East Asia affects the diurnal changes of DOFS from the CO retrievals by FY-4B/GIIRS, as shown in Figure 3(b)"**

Ln. 314.: The results of Figure 3 are only for North China Plain, but have you done it also for the two other regions? Mongolia has a more complex diurnal TC change than North China Plain but the surface pressure/topography is also different between the two regions. Would the results of Figure 3 be the same for Mongolia region or not?

In the revised manuscript, we have updated the figure and show the correlation between TC and DOFS and the retrieval accuracy from the synthetic experiments in two different figures: Figure 3 and Figure 4 in the revised manuscript, and Fig. S2, Fig. S3 and Fig. S4 in the supplementary materials. We have also added North India as the fourth region of interest. In addition, the correlation plot between TC and DOFS based on all the retrievals in July of 2022 for the four selected regions are also shown in Figure 9. Indeed, we see that Mongolia has a wider range of TC changes. As a result, its correlation with DOFS are different from the other three regions.

Ln. 315. It is confusing, the "truth" is based on the ECMWF EAC4 results but it is also used as CO a priori profile in your retrieval algorithm, so what is the difference between the comparison of (1) and (2)? How can you compare the retrievals to ECMWF

EAC4, if you already used this CO profile as a priori profile in your retrieval algorithm (see Ln. 291)?

In the revised manuscript, we have adopted a different strategy for the a priori CO profile. We have used a fixed a priori CO profile and re-made the comparison. Now the "truth" CO columns from the model simulations and the static CO a priori profile are independent of each other. From our comparison results shown in Figure 4 for North China Plain, and Figure S2-4 shows the results for Mongolia, East China Sea, and North India, respectively, we found that the accuracy of the retrieval relies heavily on the DOFS which shows large difference between daytime and nighttime.

Ln. 470-479: Talking about wildfires, during the month of July 2022, several wildfires occurred in Siberia. The transport and mixing of CO in the Northern Hemisphere might have bring CO concentrations from the Siberian fires to the regions of your study. Have you looked at that? The ECMWF EAC4 and ER5 simulation/reanalysis used in your algorithm do not include 2022 and so might not be representative of CO concentrations for the month of July 2022. It is difficult to determine if your retrievals are well representative of July 2022 consequently. It could be then interesting to have evaluation of your L2 retrievals to in situ data.

Yes, the wildfire emissions have been detected by GIIRS CO retrievals around the Siberia in the north eastern corner of our study area. In Figure 13, we compare the map of CO columns from GIIRS and IASI in the daytime and nighttime on July 07, 2022. Due to different cloud screening process, the available data points around the wildfire region may be different. Nonetheless, we can see the high CO levels in the Siberia around 60°N and 135°E that has been well captured by GIIRS and IASI. We have added the following statements in the revised manuscript: "**We can see the spatial distribution of CO columns in the daytime is characterized by two source regions, the North China Plain as a source of anthropogenic emissions and the Siberia region as a source of natural wildfire emissions. This wildfire over Siberia on July 07, 2022 is further confirmed using MODIS optical images and fire counts (not shown here). In the nighttime, the emissions from North China Plain persist. However, the wildfire regions are not covered. The scatter plots comparing the collocating observations show good agreements between the two datasets**" in Section 6.4, and "**In addition, elevated CO column values can be detected around the Siberia region which is close to the north eastern region of our study area. The high CO values are related to the wildfire emissions over the region which intensifies in the summer season**" in Section 6.3.

The model simulations from CAMS EAC4 assimilates observations from MOPPIT and IASI for CO simulations, so the wildfire emissions should in theory be assimilated in the model. However, since the most updated data are not available, a comparison is not feasible. Similarly, the most updated in situ data are not available at this moment. Therefore, we rely on the collocating IASI retrieval, as an independent data source, to compare with our retrievals.

The wildfire emissions of CO can also be seen from the averaged diurnal maps in Figure 11 and Figure 12. The high values around the region close to Siberia are mostly affected by the CO emissions from the Siberia wildfires in July.

Appendix A: Figure A1.a is not used in the study. Considering the wildfires in July 2022, was there fire CO emissions included in CAMS model?

Yes, it has assimilated wildfire emissions. However, the most updated simulations are not available at this moment. We used the same period data in 2021. In the revised manuscript, we have changed the a priori to be a fixed CO profile with a large variability in the retrieval algorithm. Through comparison with IASI, we found our results are consistent in capturing unexpected natural events such as wildfire emissions. Please also see our response to your last comment.

This study was only done for a summer month, but have you look at other season? The diurnal cycle might be associated to meteorological conditions and emissions patterns different by season. Additionally, having hourly data and comparing North China with Mongolia and East China Sea, I was wondering if you looked at the difference in CO concentration between these regions during the daytime. I would expect to observe highest concentration for North China than Mongolia during the morning time corresponding to rush hours, however this would depend on synoptic disturbances.

In the revised manuscript, we added Section 7.1 in the Discussion section and Fig. S9, Fig. S10, and Fig. S11 to discuss the preliminary results from applying the algorithm to a month in the winter season (December 2022) when the temperature of the surface and the atmosphere become with low compare to July. The results show that the spectral fitting residual increases by a factor of 2, leading to larger retrieval uncertainty and less available retrievals after applying the quality control filters. However, we see the lower latitude land the ocean region, the temperature is still high and can be a good study area using GIIRS CO retrievals in winter. For details, please see our discussions in Section 7.2

---

## Author Comment (AC3)

We thank the Dr. Mengqi Zhang for his/her constructive comments and suggestions to improve the quality and clarity of our manuscript. Item-by-item responses to the specific comments are provided below, in which the reviews' comments are in blue, our responses in **black**, and modifications of the original manuscript are indicated by highlight in yellow in the revised manuscript.

**Community comment:**

This article used the GIIRS, which is the fisrt geostationary Infrared hyperspectra Sounder over the world, to retrieve the diurnal carbon monoxide. CO is very important atmospheric pollutant and a tracer of CO2. This work is very meaningful and the paper is well-writen and well organized. This would be the sencond work on atmospheric trace gas retrieval after Lieven Clarisse(2021)(https://doi.org/10.1029/2021GL093010), and also is the fisrt work on GIIRS-FY4B and CO.

As a community comment, I highly recommend publication to raise awareness of thermal infrared detection of trace gases.

Thank you for your very positive comments.

I also have the following suggestions and questions.

The colormap in Figure 10(a) and Figure 11(b) shoud be changed. The viridis colormap is hard for reading and knowing the spatial change. May jet ,rainbow, or some other colormap are suitable.

We have changed the colormap to "Spectral" in Python which has a range of colors for describing the CO gradience in the study area.

The GIIRS of FY4A has a certain degree of wavelength calibration offset. Is the GIIRS of FY4B better in wavelength calibration? How is this considered in the inversion? Should the wavelength be calibrated first, or should it be brought into the inversion model for optimization iterations?See the GIIRS FY4A wavelength calibration problem on: https://www-cdn.eumetsat.int/files/2021-01/8%20-%20Coheur%20-%2017h15%20-%20Preliminary%20results%20on%20NH3%20retrievals%20using%20GIIRS.pdf

Thanks for bringing this up. Fortunately, FY-4B/GIIRS has a much better performance, compared to its predecessor, according to Li et al. (2022). Also from our own assessment, there is no evidence of a large spectral shift, judged from the spectral fitting residual, that may affect our retrievals.

P12,L320. The $x_a$, $x_{true}$ in the formula shoud be differentiated from the previous formula (Eq 5). The same express $x_a$ and $\mathbf{x_a}$ may lead some confusion. May CO_a or CO_true

be better. This may be helpful for some readers. **x_a** in Eq 5 including CO_a and other state vectors.

Thank you for your great suggestion. We have made the changes as suggested.

Are there any plans to apply the algorithm to FY4A with data from 2019? (Although FY4B has better instrument performance.)

Yes. We do have a plan to apply the retrieval algorithm to FY-4A/GIIRS as it has a longer time period. But it will require some re-calibration to fix the spectra shift like you have just mentioned, and re-evaluation of the spectral accuracy.

GIIRS completes a scan cycle in about 2 hours, so the data at a certain position within 0-2h is just an instantaneous value within the cycle. Although there is no difference in value, it may be better to remind readers to pay attention.

We have added this statement in Section 6.3.

The temperature profile is a key physical quantity for CO inversion, and the ERA5 reanalysis data was used in this study. How sensitive is the algorithm to the temperature profile? The CO2 absorption band of GIIRS has the ability to invert temperature profiles. Would the inversion results for trace gases be better using their inversion temperature profiles?

The atmospheric temperature profile is a key parameter as it regulates the atmospheric thermal emissions. The temperature profile from ERA5 reanalysis has been shown to be very accurate as it assimilates various observations. Nonetheless, we have retrieved a scale factor (with a priori of one and a small variability allowed) to the temperature profile. The retrieved scale factor is still very close to one, suggesting using the reanalysis is not causing a bias.

Retrieving the temperature profile from GIIRS's own spectra is an alternative way of getting the profile information for our retrieval algorithm, but it will take some time to get this information ready. This can be done in future investigations.

P5, Figure 1c. …. (bottom) Jacobian for CO ….. May add the matrix would be better( Jacobian matrix).

The Jacobian values in Fig. 1C is a vector, representing the change of radiance at different channels to a perturbation in CO concentration. So we keep "Jacobian" in the label. Instead, in the figure caption, we added "… **Jacobian at different channels …**"

Overall, this article is very valuable and meaningful. I highly recommend publication.

Thanks again for your positive comment.

---

## Author Response (AR2)

We thank the reviewer for his/her constructive comments and suggestions to improve the quality and clarity of our manuscript. Item-by-item responses to the specific comments are provided below, in which the reviews' comments are in blue, our responses in **black**, and modifications of the original manuscript are indicated by highlight in yellow.

Report #1

The authors have improved the manuscript and answered all previous questions. However, I still have three comments that need to be answered.

General comments:

1) You have changed the a priori for all retrievals to a fixed CO profile. But how do you account for difference in seasonality in your a prior profile? You mention that a fixed a prior profile might help interpreting the results, but what are the uncertainties associated with this fixed a prior profile compared to a variable a prior profile? A variable a prior profile might better capture variability and seasonality at different latitudes. Further discussion should appear regarding the use of the fixed a prior profile compared to a variable a prior profile.
It is not clear also if your a priori profile is a spatial mean of the whole region of interest (Figure 1.a) or if you have different a prior profile for each specific region over land?

The fixed a priori CO profile is assumed for different time and locations. So, there is no seasonality in the a priori profile. The uncertainty of the a priori profile is determined by the data variability (one standard deviation) from CO simulations, shown as the error bar in Figure 2(a). The a priori profile is a spatial mean of the targeted regions of interest. Although a variable a prior profile might better capture variability and seasonality at different latitudes, it may not reflect the information existed in the observed spectra.

Related statements have been added to the revised manuscript (Section 3.2.1).

2) For the comparison between IASI and GIIRS, such as in Figure 13, you adjusted the a prior CO profile with consideration of the AVK. (equation 6). You mention that the vertical sensitivities of both sensors are similar, but according to Figure A.2, they do have difference, particularly around 200 hPa. Additionally, vertical sensitivity in Mongolia peaks at different pressure between the 2 sensors. For precision and comparison, should not be better to use the averaging kernel of IASI in equation 6? As they do have some differences, why not using the AVK smoothing for the comparison between IASI and GIIRS?

Although there is systematic difference in AK around 200 hPa, the impacts on the CO columns are very small, since the CO partial column at around 200 hPa is much smaller than the lower atmosphere. Please see the result from the comparison experiment below:

To compare the smoothing effects of the averaging kernel (AK) matrix from GIIRS and IASI retrievals, we applied the AK smoothing to the IASI retrieved CO partial column profile retrievals which are assumed to be close to the truth. Following **Luo et al. (2007)**, we smooth the partial column profiles using the GIIRS or IASI AK matrix, given by:

$$x^*_{smoothed} = A^* x^{IASI}_{ret} + (I - A^*) x^{IASI}_a \tag{S1}$$

where $A^*$ is either the GIIRS or the IASI AK matrix, and $x^*_{smoothed}$ is the corresponding smoothed result. Since GIIRS and IASI have different footprint sizes and spatial resolutions, for every IASI data, the closest GIIRS AK matrix is used. The comparison results are shown in **Figure S9**. We can see the AK smoothed CO columns are high consistent, suggesting that the small difference in AK between IASI and GIIRS are not significantly affecting the comparison of the retrieved CO columns.

Related statements have been added to Section 6.4 in the revised manuscript.

(a) (b)

[Figure]

**Figure S9. Comparison of AK-smoothed CO columns between GIIRS and IASI for (a) daytime and (b) nighttime data corresponding to Figure 13. The linear fit slopes, correlation coefficients (r), and root-mean-square-error (rmse) are also indicated.**

3) ECMWF EAC4 reanalysis was used to construct a fixed CO a prior profile. It is mentioned nowhere in your paper that this reanalysis do assimilate MOPITT and IASI (which can consequently capture fire information) as precise in your author's reply. The fact that MOPITT and IASI are assimilated in the reanalysis used for constructing your CO a prior profile which is then used for GIIRS should be mentioned in your manuscript. Additionally, you compare readjusting GIIRS profiles and IASI for evaluation, but could this evaluation be biased since you are using IASI information in your GIIRS a prior profile? Please develop.

The a priori profile is a spatial mean of the targeted regions of interest, and it is static for different time and locations. So, the wildfire information will be washed out when averaging from a large number of simulated profiles with the majority not affected by wildfire emissions.

We added the following statements in the revised manuscript:
**Noted that the EAC4 has assimilated MOPITT and IASI retrievals which can capture wildfire information. However, such information has almost completely reduced in the resulted a priori profile which is averaged from a large number of simulated profiles with the majority not affected by wildfire emissions.**

Technical comments:

1) Figure 10. The caption should be corrected. The error bars are not represented in figure 10.c. Additionally, there is no figure 10.d.

The caption error has been corrected.

The authors have made substantial changes that led to large improvements of the manuscript:
- change of a diurnal varying a priori to a fixed one to document the diurnal information really contained in the measurements
- addition of boundary layer height from ECMWF and CO columns from EAC 4 reanalysis
- change from 0-1 km layer with little information to 0-3 km with more relevant information
- improved comparisons between GIIRS and IASI

These changes improve the content of the paper and better document the abilities of the GIIRS retrievals. Nevertheless, the abstract and conclusions have not been changed accordingly.

Abstract :
The last sentence « This study demonstrates the capability of GIIRS in observing the diurnal CO changes in East Asia. » is misleading and does not really correspond to what is demonstrated in the paper. Indeed Fig. 10 shows that GIIRS enable the detection of the absence of diurnal variability in the CO total column over large East Asian regions but the sentence makes the reader believe that it enables to detect the expected ground level CO diurnal variations linked to emission diurnal variations. I suggest to mitigate this statement as follows:
« This study demonstrates that GIIRS correctly reproduces the low diurnal variability of CO total columns over large East Asian regions. Nevertheless, the GIIRS retrievals does not enable to detect the larger ground level CO diurnal variability linked to surface emissions changes. The CO retrievals are indeed impacted by BLH and information content diurnal changes entangled with the CO ground level variability. »

We changed the last sentence to:
**"This study demonstrates that the GIIRS retrievals are able to reproduces the temporal variability of CO total columns over East Asia in the daytime in July. Nevertheless, the retrievals have low detectivity in the nighttime due to their weak sensitivity to the ground level CO changes limited by low information content. Model assimilation that takes into account the retrieved diurnal CO profiles and the associated vertical sensitivity will have potential in improving local and global air quality and climate research over East Asia."**

Conclusion :
The corrections and mitigations made to the paper should also be included in the conclusion.
The conclusion has to mention the unability of the GIIRS measurements to detect the diurnal variability of surface CO linked to emission variations over industrialized regions because of BLH and DOFS diurnal variations (section 6.3).
Specifically, the sentence « These results demonstrate the capability of GIIRS in constraining the diurnal CO changes in East Asia. » is misleading and rather an overstatement. It makes the reader believe that GIIRS is able to constrain the CO changes close to the ground and therefore the emissions which is not the case as demonstrated by the paper. This statement should therefore be mitigated as proposed for the abstract.

We have made the following changes:
**"This study demonstrates that the GIIRS retrievals are able to reproduces the temporal variability of CO total columns over East Asia in the daytime in July. Nevertheless, the retrievals have low detectivity in the nighttime due to their weak sensitivity to the ground level CO changes limited by low information content.** Since CO plays an important role in tropospheric atmospheric chemistry and is an effective tracer of $CO_2$, the CO **profile** retrievals at a spatial resolution of 12 km and a temporal resolution of 2 hours from GIIRS have great potential in improving local and global air quality and climate research **through model assimilation that takes into account the associated vertical sensitivity."**

Minor comments :
There are a number of typing and syntax errors that have to be corrected.
L371 : « solar zenith angles less than 70° » : this means only part of the daytime measurements but
the results show 24 hrs results.
We changed "solar zenith angles" to "viewing zenith angles"

Fig 10 : « the error bars for ( c) .. »
The caption error has been corrected.